# Atomic layer deposition triggered Fe-In-S cluster and gradient energy band in ZnInS photoanode for improved oxygen evolution reaction

Linxing Meng[1], Jinlu He[2], Xiaolong Zhou[3], Kaimo Deng[1], Weiwei Xu[1], Pinit Kidkhunthod[4], Run Long[2], Yongbing Tang [3] & Liang Li[1✉]

Vast bulk recombination of photo-generated carriers and sluggish surface oxygen evolution reaction (OER) kinetics severely hinder the development of photoelectrochemical water splitting. Herein, through constructing a vertically ordered ZnInS nanosheet array with an interior gradient energy band as photoanode, the bulk recombination of photogenerated carriers decreases greatly. We use the atomic layer deposition technology to introduce Fe-In-S clusters into the surface of photoanode. First-principles calculations and comprehensive characterizations indicate that these clusters effectively lower the electrochemical reaction barrier on the photoanode surface and promote the surface OER reaction kinetics through precisely affecting the second and third steps (forming processes of O* and OOH*) of the four-electron reaction. As a result, the optimal photoanode exhibits the high performance with a significantly enhanced photocurrent of 5.35 mA cm$^{-2}$ at 1.23 V$_{RHE}$ and onset potential of 0.09 V$_{RHE}$. Present results demonstrate a robust platform for controllable surface modification, nanofabrication, and carrier transport.

[1] School of Physical Science and Technology, Jiangsu Key Laboratory of Thin Films, Center for Energy Conversion Materials & Physics (CECMP), Soochow University, Suzhou, P. R. China. [2] College of Chemistry, Key Laboratory of Theoretical & Computational Photochemistry of Ministry of Education, Beijing Normal University, Beijing, P. R. China. [3] Functional Thin Films Research Center, Shenzhen Institutes of Advanced Technology, Chinese Academy of Sciences, Shenzhen, P. R. China. [4] Synchrotron Light Research Institute, Nakhon Ratchasima, Thailand. ✉email: lli@suda.edu.cn

Photoelectrochemical (PEC) water splitting can convert solar light into hydrogen energy, providing a promising path to solve the energy crisis and environmental pollution[1,2]. Zn–In–S-based ternary chalcogenide semiconductors have been widely applied as photocatalysts and PEC photoelectrodes owing to their efficient optical absorption in the visible-light region together with superior stability over binary chalcogenides[3–5]. Nevertheless, the serious recombination of photogenerated charge carriers and dull surface oxygen evolution reaction (OER) kinetics limit their application[6]. The bulk recombination of photoelectrodes can be alleviated by doping[7,8], morphology[9,10], and heterojunction engineering[11–13]. Among these strategies, constructing photoelectrodes with vertically ordered morphology is facile and effective[14,15]. However, there is usually a disordered layer between the substrate and ordered ZnIn$_2$S$_4$ (ZIS) nanostructures, which lowers the bulk separation efficiency ($\eta_{sep}$) of the photoanode[16–18]. As another major factor limiting the PEC performance, the OER involving four-electron reaction is dynamically slow on account of required two steps of O–H bond cleavage and O–O bond generation[19–21]. The complicated OER processes have a high reaction barrier, which are described by Eqs. (1)–(4)[22]. The symbol * marks the activation site of photoanode, and OH*, O*, and OOH* indicate the reaction intermediate groups:

$$* + H_2O \rightarrow OH^* + e^- + H^+ \tag{1}$$

$$OH^* \rightarrow O^* + e^- + H^+ \tag{2}$$

$$H_2O + O^* \rightarrow OOH^* + e^- + H^+ \tag{3}$$

$$OOH^* \rightarrow * + O_2 + e^- + H^+ \tag{4}$$

Various cocatalysts (Ni/FeOOH[23–25], CoO$_x$[26], and layered double hydroxide (LDH)[27], etc.[28–30]) have been designed to enhance OER kinetics. These cocatalysts may lead to the decreased light-harvesting capability and increased charge recombination rate of photoanodes, owing to the thickness of cocatalyst layer and additional interface defects introduced between cocatalysts and photoanodes[31]. Alternatively, the in situ

grown bonding-effect-based catalytic groups, such as oxysulfide photocatalyst[32,33], Zn–O–Co–O–Zn configuration[34], Ir–O–V groups[35], and Fe–O–Ni bridge[19], et al.[36,37], can tackle the above problems. They can provide catalytic sites for water dissociation and adjust the adsorption energies of water molecules and intermediates[35]. This efficient strategy mostly acts on the entire bulky electrode in the water electrolysis. For the PEC, more attention should be paid to the OER reactions that take place on the photoanode surface[38]. It is still challenging to generate cocatalyst groups only on the photoanode surface without affecting the bulk to improve OER reaction kinetics.

To address the above critical issues, the atomic layer deposition (ALD) technique is used to introduce Fe and O atoms into two-dimensional Zn$_{10}$In$_{16}$S$_{34}$ (ZISZ) ordered nanosheet arrays (denoted as ZISZ/Fe), where the Fe–In–S clusters are formed on the surface while the Zn–O bond is formed at bottom of the ZISZ photoanode. The Fe–In–S clusters greatly reduce the surface overpotential ($\eta$) and interfacial recombination, and thus boost the kinetics of OER through precisely affecting the forming processes of O* and OOH*. The change of element content and species along the cross-section of ZISZ causes the gradient energy level alignment within the photoanode, together with the uniformly ordered morphology without disordered layer, promoting the $\eta_{sep}$ in the bulk. The ZISZ/Fe photoanode shows a largely increased photocurrent ($J$) of 5.35 mA cm$^{-2}$ at 1.23 V$_{RHE}$ and onset potential ($V_{on}$, the potential at which 0.02 mA cm$^{-2}$ current density was first measured) of 0.08 V$_{RHE}$, which is 32 and 6.3 times higher than that of ZIS and ZISZ. The performance is comparable to and even better than other up-to-date reported photoanodes-based sulfides (Supplementary Table 1).

## Results and discussion

**Preparation and characterization of photoanodes.** Figure 1a displays the fabrication process of photoanodes with different morphology through controlling the solvent of precursors. The scanning electron microscopy (SEM) images of ZIS, ZISZ, and ZISZ/Fe nanosheet arrays are shown in Fig. 1b–g. They have the same morphology except that a lot of disordered small nanosheets

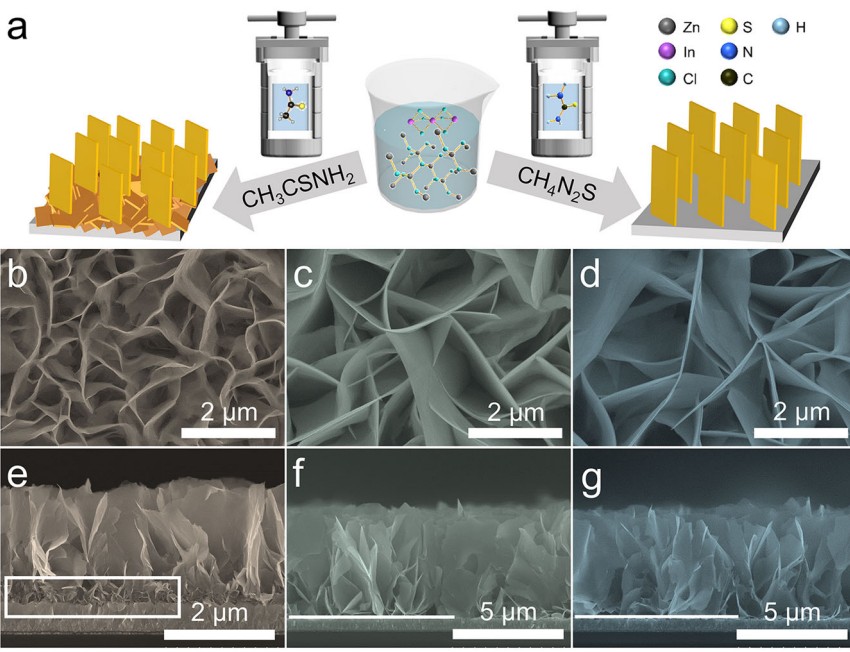

**Fig. 1 Synthesis and morphology comparison of photoanodes. a** The schematic diagram for the synthetic process of the photoanode. **b–d** Top-view and (**e–g**) cross-sectional SEM images of ZIS, ZISZ, and ZISZ/Fe nanosheet arrays.

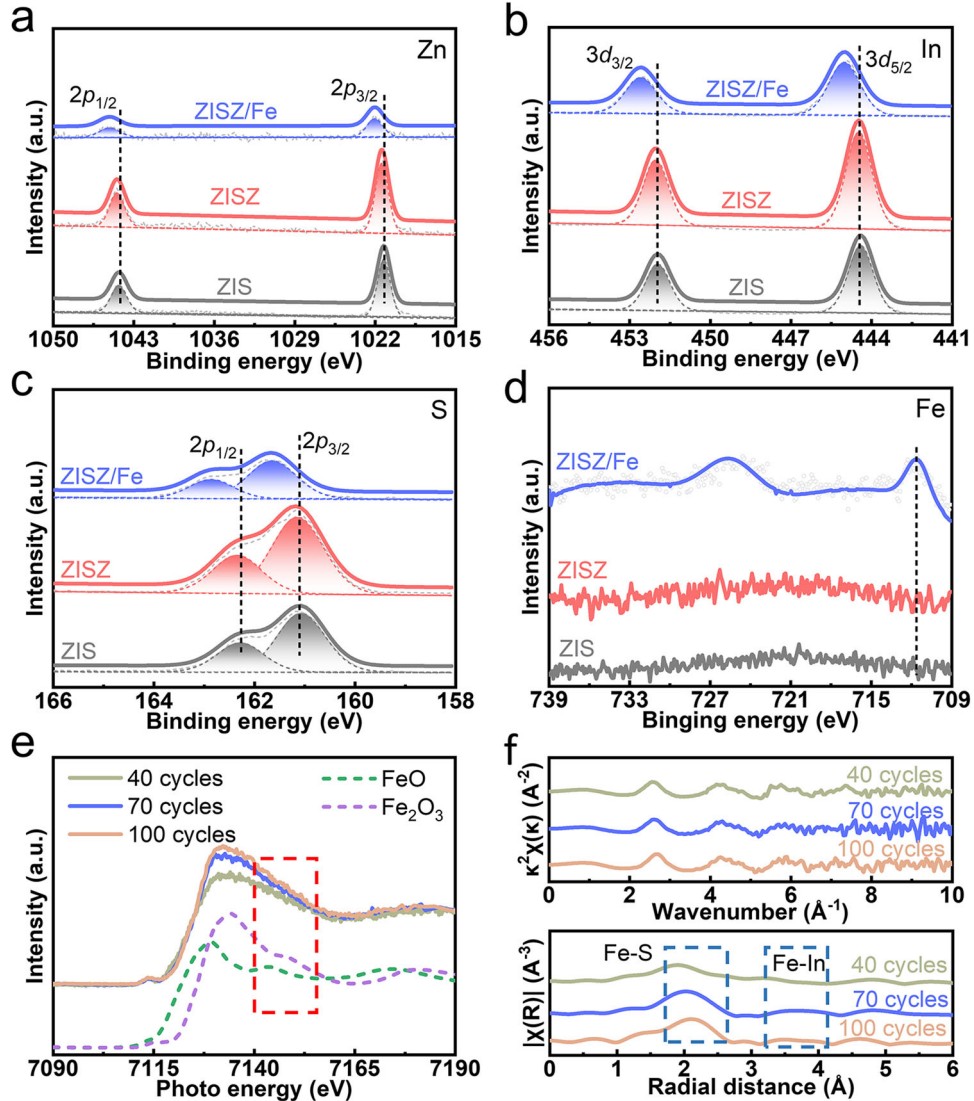

**Fig. 2 Characterizations of photoanodes. a** Zn 2*p*, **b** In 3*d*, **c** S 2*p*, and **d** Fe 2*p* XPS of ZIS, ZISZ, and ZISZ/Fe. **e** The Fe K-edge XANES spectra. **f** The Fe K-edge EXAFS as functions $k^2\chi(k)$ and FT-EXAFS of ZISZ/Fe with different ALD cycles (40, 70, and 100 cycles); k = wave vector and $\chi(k)$ = oscillation as a function of the photoelectron wavenumber.

are at the bottom of ZIS sample. The transmission electron microscopy (TEM) images (Supplementary Fig. 1) further reveal a typical nanosheet morphology. The vertically ordered two-dimensional nanosheets provide a direct transport path for photogenerated carriers, thereby reducing the bulk recombination of photoanode. The phase of the samples was determined by X-ray diffraction (XRD) pattern (Supplementary Fig. 2). ZIS reveals the typical hexagonal $ZnIn_2S_4$ (JCPDS: 72-0773)[18]. ZISZ exhibits a characteristic peak of $Zn_{10}In_{16}S_{34}$ (JCPDS: 27-0989)[39]. When the ZISZ is treated by ALD, its peak position does not shift obviously. The corresponding high-resolution TEM (HRTEM, Supplementary Fig. 3) and selected area electron diffraction (SAED, inset in Supplementary Fig. 3) results show that all the samples have similar crystal structure, and the crystallinity of ZISZ and ZISZ/Fe is improved. The energy dispersive spectroscopy (EDS) elemental mapping of ZISZ/Fe in Supplementary Fig. 4a–f reveals that Zn, In, S, Fe, and O elements are uniformly distributed across the entire nanosheets.

The X-ray photoelectron spectroscopy (XPS) spectrum was conducted to characterize the surface chemical environment of elements (Fig. 2a–d). Zn, In, and S peak position of ZISZ change

slightly compared with ZIS, indicating that they have similar chemical composition and state. Once the Fe and O elements are introduced by ALD treatment, the characteristic Fe peak appears (Fig. 2d). The Zn, In, and S peak position of ZISZ/Fe shifts obviously toward higher energy and the intensity is reduced greatly. This shift may be due to the formation of Fe–In–S clusters or the increased number of O elements with strong electronegativity, which will be discussed later. Supplementary Fig. 5 shows that the lattice O is not formed, and the peak intensity of O related with the surface-absorbed $H_2O$ (531.81 eV) and the OH group (529.86 eV) is enhanced because the $O_3$ treatment increases the hydrophilicity of surface during the ALD process[38]. The reduced intensity of In results from the replacement of Fe to build Fe–In–S cluster. This is further confirmed by XPS of samples treated under different ALD cycles (Supplementary Fig. 6). With the increased cycles, the peak intensity of In gradually decreases while that of Fe increases. Furthermore, peaks located at 712.92 and 710.86 eV correspond to $Fe^{3+}$ and $Fe^{2+}$, respectively, indicating the chemical state of substituted Fe changes with increased Fe[40]. By performing DFT calculations using the models shown in Supplementary Fig. 7, we

find that substitution of an In with an Fe requires smaller formation energy than replacement of a Zn with an Fe, which suggests that the former substitution is energetically favorable. The calculated Bader charge and magnetic moment on the Fe ion correspond to 1.687 and 2.662, indicating that the oxidation state and spin state of the Fe in the ZISZ/Fe system are +2 and 3. Furthermore, the replaced Fe atom is distant from the Zn atom and facilitates to form Fe–In–S bond between the Fe dopant and its surrounding In and S atoms.

In order to reveal the state of Fe in the whole ZISZ/Fe, XPS at different depths through Ar ion etching was measured (Supplementary Fig. 8). As the etching depth increases, the peak intensity of Zn gradually increases and its peak position shifts toward higher energy (Supplementary Fig. 8a). The peak intensity of In also increases while the peak position shifts toward lower energy (Supplementary Fig. 8b). When the etching depth increases to 50 nm, the signal of Fe substantially disappears (Supplementary Fig. 8c), but the peak position of O shifts toward lower energy (Supplementary Fig. 8d). According to these data, we can get the following conclusions: (1) Fe is mainly deposited on the surface with a thickness of about 50 nm, which may be due to the limited diffusion rate of Fe in the ALD chamber. (2) The Zn peak shifts toward higher energy and O peak shifts toward lower energy is due to the formed Zn–O. The Zn–O bond is shorter than Zn–S bond, thus the binding energy of Zn–O is larger. (3) The opposite trend of the peak intensity of In and Fe also indicates the successful substitution. To further convince the construction of Fe–In–S clusters on the photoanode surface, X-ray absorption spectroscopy (XAS) was used to survey the Fe $L_3$ absorption edges. As observed in Fig. 2e, the K-edge dotted lines located at around 7090 and 7190 eV represent the standard curves of FeO and $Fe_2O_3$, respectively. The peak region (7110–7170 eV) suggests that the photoanode contains different Fe content. Unlike $Fe_2O_3$ standard sample, there is no feature at the energy range between 7140 and 7165 eV of our prepared samples. The different characteristics of curves from standard FeO and $Fe_2O_3$ samples manifests that no oxides are formed. Furthermore, Fig. 2f shows the Fe K-edge extended X-ray absorption fine structure (EXAFS) as functions $k^2\chi(k)$ and its Fourier-transformation (FT-EXAFS) (FT-$\kappa^2\chi(\kappa)$) for ZISZ/Fe prepared with different ALD cycles. The peaks at 2.1 and 3.5 Å are assigned to Fe–S bond and Fe–In bond, respectively, and the content of Fe–S bond increases with the increased deposition cycles. These characterizations indicate that the Fe–In–S clusters are successfully formed on the surface of photoanode.

**PEC performance and characterizations for mechanism.** Supplementary Fig. 9 displays the SEM images of ZISZ/Fe samples prepared with different ALD cycles. The surface morphology has no apparent difference owing to the slow growth rate (about 0.05 Å per cycle) of Fe during the ALD process. The photocurrent–voltage (J–V) curves of samples indicate the J increases first and then decreases, and the optimal performance appears in the 70th cycle (Supplementary Fig. 10). Compared to ZIS, the performance of ZISZ and ZISZ/Fe photoanodes is significantly improved (Fig. 3a), and the J reaches 0.84 and 5.35 mA cm$^{-2}$ at 1.23 V vs. RHE, respectively. the $V_{on}$ is related to photovoltage and surface OER barriers, and the photovoltage can be determined by the open-circuit potential (OCP) measurement. The greater photovoltage and reduced surface OER barriers can cause the cathodic shift of $V_{on}$ in photoanode. The $V_{on}$ of ZISZ has a slightly positive shift compared with ZIS, as is also revealed by OCP (Supplementary Fig. 11), that ZISZ possesses the smallest value. Although the OCP of ZISZ/Fe is still smaller than ZIS, it possesses a cathodic $V_{on}$, because the

existence of Fe–In–S cluster reduces the surface electrochemical reaction barrier of ZISZ/Fe photoanode[41,42]. The applied bias photon-to-current efficiency ($\eta_{ABPE}$) of photoanode demonstrates the evident performance improvement of ZISZ/Fe under different bias voltages (Supplementary Fig. 12). The capability of light harvesting gradually increases with the decreased bandgap for ZIS, ZISZ, and ZISZ/Fe (Fig. 3b). The incident photo-to-current efficiency (IPCE) and absorbed photon-to-current efficiency (APCE) show the increased IPCE and APCE value over the whole wavelength range (Supplementary Fig. 13), indicating the outstanding PEC performance of ZISZ/Fe. The IPCE and APCE are in accordance with the variation of photocurrents shown in Fig. 3a. This superior performance confirms the excellent light absorption, $\eta_{sep}$, and injection efficiency ($\eta_{inj}$), which we will discuss as follows. The $\eta_{sep}$ and $\eta_{inj}$ were evaluated using 0.25 M $Na_2SO_3$ + 0.25 M $Na_2S$ as a hole sacrificial agent (Supplementary Fig. 14). The $\eta_{sep}$ of ZISZ is higher than that of ZIS (Fig. 3c), mostly thanks to the vertically ordered nanosheet distribution at the bottom as efficient transport path. The little improvement in $\eta_{inj}$ (Fig. 3d) also signs that this ordered morphology mainly raises the $\eta_{sep}$. To investigate the bulk carrier dynamics, room-temperature photoluminescence (PL) was conducted (Supplementary Fig. 15). The PL intensity of ZISZ is lower than that of ZIS, suggesting the faster separation of electron-hole pairs and reduced bulk recombination in the ZISZ. When Fe is introduced into the sample by ALD, both the $\eta_{sep}$ and $\eta_{inj}$ are improved evidently. The enhanced $\eta_{sep}$ is ascribed to the formation of lattice oxygen and the modification of the energy band, which will be discussed in detail below. The improved $\eta_{inj}$ of ZISZ/Fe confirms the role of Fe–In–S cluster played in improving the surface OER dynamics. To rationalize the improved catalytic performance of the ZISZ/Fe photoanode, we obtained the $\eta$ for OER of the ZISZ and ZISZ/Fe systems based on the DFT calculated free energy changes (Supplementary Table 2) using the two structures shown in Supplementary Fig. 7a and e. The computational details are described in Supplementary Material. The $\eta$ decreases to 2.499 V in the ZISZ/Fe from 3.522 V in the pristine ZISZ (Fig. 3e and f). The notable 1.023 V drop in $\eta$ provides a solid evidence that the catalytic performance is enhanced in the ZISZ/Fe photoanode once the Fe–In–S clusters are formed, because the Fe dopant alters the rate-determining step of *O to *OOH (step 3) in the ZISZ system (Fig. 3e) into *OH to *O (step 2) in the ZISZ/Fe (Fig. 3f) system, leading to that the OOH* is preferred for formation on the ZISZ/Fe catalyst and favors $O_2$ production. The DFT calculations provide an excellent explanation for the experimental results. The electrochemical impedance spectroscopy (EIS) was further measured to probe the interface reaction dynamics of electrode (Supplementary Fig. 16). The series resistance ($R_s$) represents the resistance between FTO and sample, and the charge transfer resistance ($R_{ct}$) indicates the resistance between photoanode and electrolyte[22]. ZISZ and ZIS perform the similar $R_{ct}$, further demonstrating that the activity improvement of ZISZ is attributed to the reduced bulk recombination instead of interface charge transfer. ZISZ/Fe possesses the smallest $R_{ct}$, suggesting that the enhanced surface OER reaction kinetics is caused by Fe–In–S clusters. In order to prove that the promotion of surface OER is not related to O, only $O_3$ was introduced into the ALD chamber (samples are denoted as ZISZ/O) to avoid the formation of Fe–In–S. The J of ZISZ/O is lower (Supplementary Fig. 17a) and the surface resistance is higher (Supplementary Fig. 17b), therefore, the Fe–In–S clusters are critical for enhanced OER reaction.

**Charge transfer and recombination kinetics.** Intensity-modulated photocurrent spectroscopy (IMPS) kinetic analysis

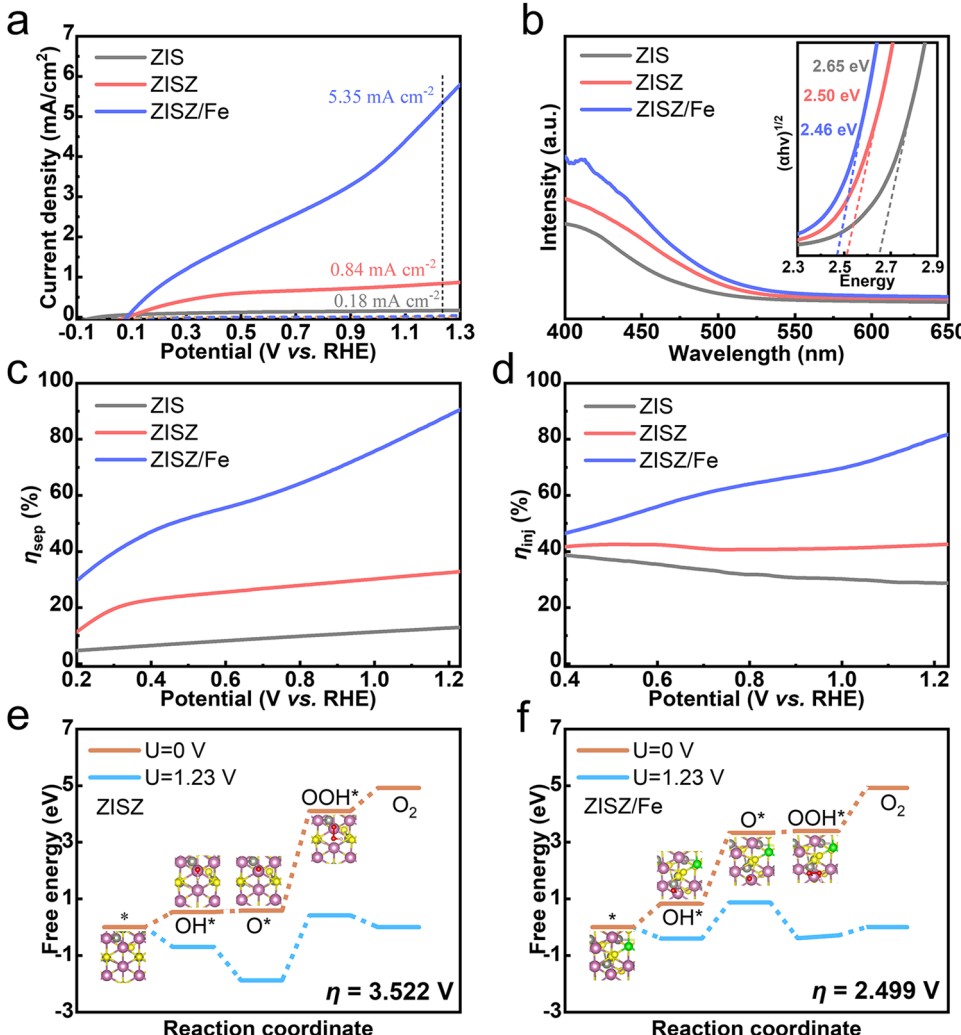

**Fig. 3 PEC performance, optical, and electrochemical characterizations. a** Linear sweep voltammogram curves. **b** UV-visible absorption spectra (the inset is the corresponding Tauc-plots). **c** The $\eta_{sep}$ and **d** the $\eta_{inj}$ of ZIS, ZISZ, and ZISZ/Fe. Free energies of OER reaction steps for **e** ZISZ and **f** ZISZ/Fe systems, the entire structure of the OER four-step reaction are shown in Supplementary Fig. 19.

was performed to study the charge transfer and transit dynamics processes, including surface charge-transfer efficiency ($\eta_{tran}$), surface charge-transfer rate constants ($K_{tran}$), and surface charge recombination rate constants ($K_{rec}$). The relationship between them can be described by the following formula[43]:

$$\eta_{tran} = K_{tran}/(K_{tran} + K_{rec}) \qquad (5)$$

The IMPS curves of ZIS, ZISZ, and ZISZ/Fe were measured with different voltages (0.6–1.2 V) under $\lambda = 365$ nm irradiation (Supplementary Fig. 18). The calculation details of the rate and time constants are presented in the supporting information. The $\eta_{tran}$ of ZISZ are basically unchanged while ZISZ/Fe has a huge improvement compared with ZIS (Fig. 4a), in accordance with the trend of $\eta_{inj}$. This confirms that IMPS is feasible to analyze surface OER reaction kinetics[38]. The values of the $K_{tran}$ over the entire voltage range are much larger than those of the $K_{rec}$ (Fig. 4b and c), indicating that most of the carriers on the photoanode surface are effectively transferred into the electrolyte, which is consistent with the $\eta_{tran}$. With the increased voltage, the $K_{tran}$ of the three samples increases obviously only in the low voltage range, and then keeps unchanged basically, while the $K_{rec}$ remains almost constant. It is suggested that the improvement in the performance of ZISZ/Fe at a lower voltage is mainly due to the increase of the transfer rate of photogenerated carriers on the

photoanode surface, proving that Fe–In–S clusters could effectively improve its interface transfer efficiency and promote the OER reaction kinetics on the photoanode surface. The $\eta_{tran}$, $K_{tran}$, and $K_{rec}$ of ZIS and ZISZ have no large difference across the entire voltage range, indicating the improved PEC performance of ZISZ is mainly ascribed to the increased $\eta_{sep}$ once again. Compared with ZISZ, both the $\eta_{tran}$ and $K_{tran}$ of ZISZ/Fe are significantly improved, while the $K_{rec}$ is obviously reduced. This is because the construction of Fe–In–S clusters on the surface of the photoanode effectively reduces the electrochemical barrier of the photoanode surface reaction, promotes the surface OER reaction kinetics, and reduces interfacial recombination, so that ZISZ/Fe exhibits the best PEC activity. In addition, electron transfer time ($\tau_d$) of photoanodes can be calculated from the formula:

$$\tau_d = \frac{1}{2 \times 3.14 \times f_{min}} \qquad (6)$$

where the $f_{min}$ is the frequency of the lowest point of the imaginary part in IMPS. The IMPS results show longer $\tau_d$ of ZISZ/Fe owing to the frequency at the minimum imaginary part of ZISZ is higher than ZISZ/Fe (Supplementary Fig. 18), which is consistent with the reduced bulk recombination. The time-resolved transient photoluminescence decay (TRPL, Fig. 4d) spectra were measured to calculate carrier lifetime also. ZISZ/Fe

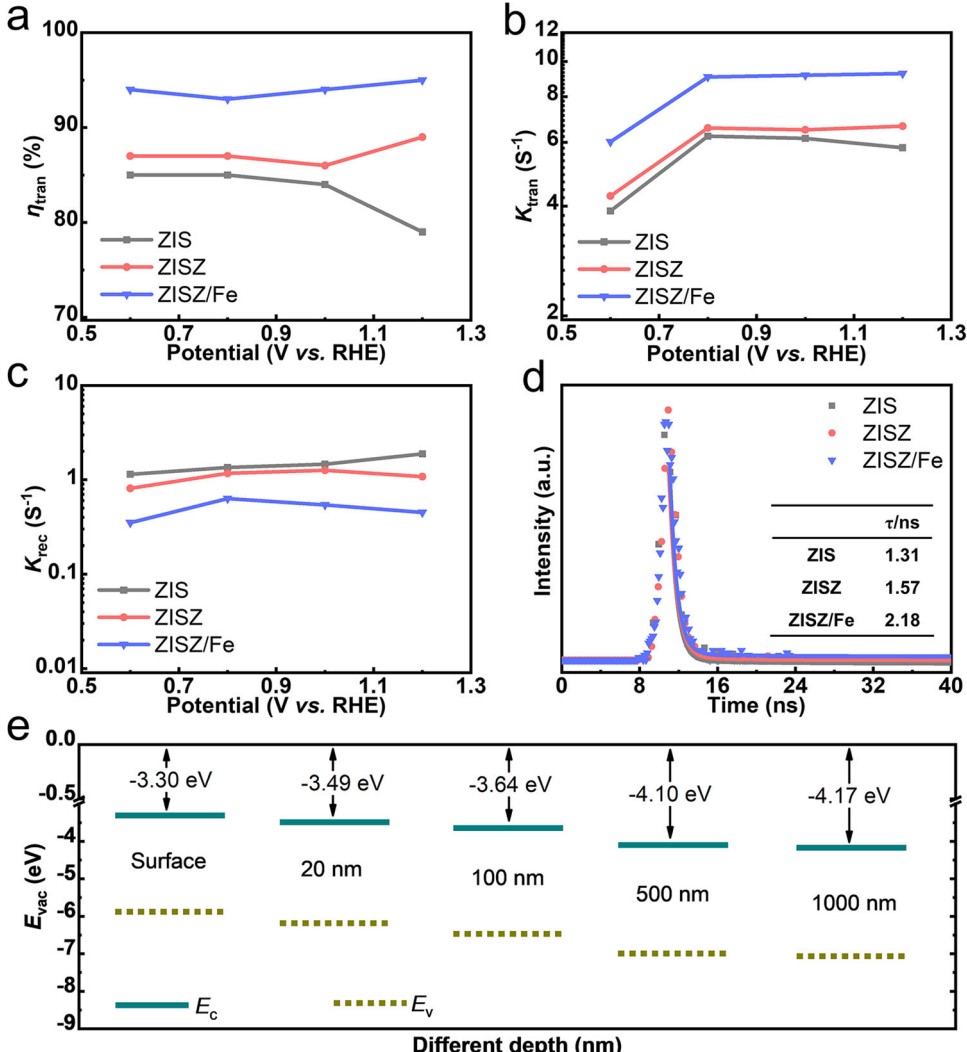

**Fig. 4 Charge transfer kinetics and energy band structure. a** The $\eta_{\text{tran}}$. **b** The $K_{\text{tran}}$. **c** The $K_{\text{rec}}$. **d** TRPL decay spectra of ZIS, ZISZ, and ZISZ/Fe (the inset is calculated carrier lifetime). **e** The energy level diagram of ZISZ/Fe with different etching depth obtained from UPS results ($E_c$: conduction band; $E_v$: valence band).

($\tau = 2.18$ ns) and ZISZ ($\tau = 1.57$ ns) display a slower recombination rate and the carrier lifetime is longer than ZIS ($\tau = 1.31$ ns). The prolonged lifetime manifests enhanced carrier transfer efficiency and decreased recombination. The ultraviolet photoemission spectroscopy (UPS) results show that the ZISZ/Fe possesses a gradient band structure throughout the nanosheets with the etching depth. The infinite number of Type-II heterojunctions causes the stepped distribution of energy level, further enhancing the $\eta_{\text{sep}}$ of ZISZ/Fe.

In summary, we developed the ALD technology to manipulate the surface chemical state and bulk energy band structure in ZnInS nanosheet ordered arrays, which are utilized as efficient photoanodes of PEC water splitting. The performance improvement is mainly ascribed to the following points: (1) The disordered part at the bottom of nanosheet arrays is eliminated and thus a direct transport channel of photogenerated carriers is formed, together with a gradient energy band structure, enhancing the $\eta_{\text{sep}}$. (2) The surface-rich Fe–In–S clusters greatly reduce the OER reaction barrier through precisely controlling the forming processes of O* and OOH* of the OER four-electron reactions. These effects largely improve the PEC activity of ZnInS photoanode with a $J$ of 5.35 mA cm$^{-2}$ at 1.23 $V_{\text{RHE}}$ and negative $V_{\text{on}}$ of 0.09 $V_{\text{RHE}}$. The present results prove the feasibility of ALD to manipulate chemical bonds and energy band structure of photoelectrodes towards improved energy conversion technology.

## Methods

**Synthesis of ZnIn$_2$S$_4$ (ZIS) nanosheet array**. The fluorine-doped tin oxide glass (FTO) was first sonicated three times with acetone, ethanol, and deionized water for 10 min each time. The ZIS precursor solution was prepared through the following steps: 0.2045 g Zinc dichloride (ZnCl$_2$) and 0.8795 g Indium chloride tetrahydrate (InCl$_3$·4H$_2$O) were dissolved into 150 mL of ultrapure water (H$_2$O), and then 0.4565 g Thioacetamide (CH$_3$CSNH$_2$, TAA) was added and stirred for 15 min. The cleaned FTO was placed in a 25 ml Teflon-lined stainless steel autoclave, with the conductive side of the FTO facing down. Each lining was pipetted with 10 mL of the prepared precursor solution, and then the vessel was sealed and heated up to 160 °C for 360 min. After cooling down to room temperature, the ZIS nanosheet was formed and rinsed with absolute ethanol, and stored in a vacuum oven.

**Synthesis of Zn$_{10}$In$_{16}$S$_{34}$ (ZISZ) nanosheet array**. The reaction process is basically the same as that of obtaining the ZIS, except for the solution of hydrothermal reaction. the ZISZ precursor solution was prepared through the following steps: 0.2045 g ZnCl$_2$ and 0.8795 g InCl$_3$·4H$_2$O were dissolved in 150 mL of H$_2$O, and then 0.6 g Thiourea (CH$_4$N$_2$S) was added and stirred for 15 min. The remaining steps are the same as the synthesis steps of ZIS nanosheet array.

**Loading Fe-In-S clusters onto ZISZ (ZISZ/Fe) nanosheet array**. Fe–In–S clusters were deposited on the surface of ZISZ by atomic layer deposition method (ALD) (LabNano 9100, ENSURE NANOTECH, Beijing, China). Ferrocene and

ozone were used as iron and oxygen precursors, respectively. The growth rate of $Fe_2O_3$ was 0.05 Å per cycle at 250 °C based on the data analysis from silicon substrates, the pulse time of both $O_3$ and Ferrocene was 0.02 s, and the interval time was 8 s. Fe–In–S cluster with different thicknesses were obtained by controlling cycle numbers.

**Materials characterization.** Scanning electron microscopy (FE-SEM, SU8100, Hitachi) was used to characterize the morphology of samples. The microstructure and lattice structures of materials were observed by transmission electron microscopy (TEM) and high-resolution TEM (HRTEM, Tecnai G2 F20 200 KV, FEI). X-ray diffraction (XRD) patterns (Cu Kα radiation, $\lambda = 0.15418$ nm, D/MAX-III-B-40KV) were obtained to evaluate the phase of the samples. The element distribution was conducted by energy-dispersive X-ray spectroscopy (EDX) under the TEM with an annular dark-field (ADF) detector. X-ray photoelectron spectroscopy (XPS) measurement (Thermo Fisher Scientific, ESCALAB 250Xi) was performed to determine the surface chemical environment of elements. X-ray absorption spectroscopy (XAS) measurements were studied in both X-ray absorption near edge structure (XANES) for Fe K-edge and Extended X-ray absorption fine structure (EXAFS) regions for Fe K-edge. The XAS experiments were carried out at the SUT-NANOTEC-SLRI XAS Beamline (BL5.2) at the Synchrotron Light Research Institute (SLRI), Nakhon Ratchasima, Thailand. The measurements were collected in transmission mode for both S K-edge and Fe $L_3$ absorption edge. Ionization chambers were installed in front of and behind the sample to detect the incident (I0) and transmitted (I1) beam. Data analysis was carried out by IFEFFIT. In order to obtain the oxidation state of S, In, and Fe, the XANES spectra were analyzed by Athena software. The UV-visible absorption spectra were recorded to characterize the light absorption of the sample by an UV-vis spectrophotometer (Shimadzu, UV-3600). The room photoluminescence (PL) spectra (excited by 325 nm illumination) were record on a RenishawRM3000 Micro-Raman system, and the excitation pulse came from a 325 W Xe lamp (Edinburgh Instruments, FLS920). The time-resolved transient photoluminescence (TRPL) decay spectra were measured by using an Edinburgh Instruments as LifeSpec II with 373 nm laser (2 Bain Square, Kirkton Campus, and Livingston EH54 7DQ). The ultraviolet photoemission spectroscopy (UPS, Thermo Scientific, Escalab 250Xi) was used to evaluate the energy band.

**Photoelectrochemical (PEC) and electrochemical measurements.** All the measurements were carried out on an electrochemical workstation (Autolab, PGSTAT 302N) with a conventional three-electrode cell, and the electrolyte was 0.5 M $Na_2SO_4$ with pH ≈ 6.8. The photocurrent–voltage ($J$–$V$) curves were performed under AM 1.5G illumination (100 mW cm$^{-2}$) from a solar light simulator (Newport, 94043A). Electrochemical impedance spectra (EIS) were measured on the workstation with the frequency ranging from 0.1 Hz to 100 kHz at an open-circuit voltage under light illumination.

The measured potential versus Ag/AgCl was converted to the reversible hydrogen electrode (RHE) scale using the Nernst equation[29]

$$E_{RHE} = E_{Ag/AgCl} + 0.059\,pH + E^o_{Ag/AgCl} \quad (7)$$

where $E^o_{Ag/AgCl} = 0.1976$ V at 25 °C and $E_{Ag/AgCl}$ is the experimentally measured potential versus Ag/AgCl reference.

The intensity-modulated photocurrent spectroscopy (IMPS) kinetic analysis was measured on an electrochemical workstation (CIMPS, Zennium Zahner) in a three-electrode system as that of the PEC measurements. The irradiation source is a LED (s/n Ls 1272, Zennium Zahner) with 365 nm wavelength, and the power intensity is 30 W m$^{-2}$. The frequency range is 1 kHz–0.1 Hz, and the ±5% modulation intensity is around 30 W m$^{-2}$. The Nyquist plots were used to get the rate constants of the surface charge-transfer and recombination[43].

The applied bias photon-to-current efficiency ($\eta_{ABPE}$) was calculated from the current-potentiometry data using the following equation[5]

$$\eta = \frac{(J_{light} - J_{dark})(mA \times cm^{-2}) \times (1.23 V_{RHE})(V)}{P_{sunlight}(mW \times cm^2)} \times 100\% \quad (8)$$

where $V_{RHE}$ is the potential of the working electrode versus the reversible hydrogen electrode, $J_{dark}$ and $J_{light}$ are the measured current density in dark and under illumination, respectively, and $P_{sunlight}$ is 100 mW cm$^{-2}$.

During the testing of injection efficiency ($\eta_{inj}$) and charge separation efficiency ($\eta_{sep}$), we employed $Na_2S/Na_2SO_3$ electrolyte as the hole scavenger. Here, $\eta_{sep}$ represents the proportion of the electrode/electrolyte interface photo-generated holes, while $\eta_{inj}$ represents the proportion of those holes at the photoanode/electrolyte interface for water oxidation[5]

$$\eta_{inj} = \frac{J_{H_2O}}{J_{Na_2SO_3}} \quad (9)$$

$$\eta_{sep} = \frac{J_{Na_2SO_3}}{J_{abs}} \quad (10)$$

where $J_{abs}$ represents the unity converted photocurrent density that is achievable, $J_{H_2O}$ and $J_{Na_2SO_3}$ represent the photocurrent density tested in 0.5 M $Na_2SO_4$ and $Na_2S/Na_2SO_3$, respectively.

The unity converted photocurrent density ($J_{abs}$) is calculated by integrating $\eta_{abs}$ over the standard solar spectrum:

$$J_{abs} = \int_{300}^{\lambda_{max}} \frac{\lambda \times \eta_{abs}(\lambda) \times E(\lambda)}{1240} d(\lambda) \quad (11)$$

where $\lambda_{max}$ is the maximum light absorption edge of a photoelectrode, $\lambda$ (nm) is the light wavelength, and $E(\lambda)$ is the power density (mW cm$^{-2}$) at a specific wavelength ($\lambda$) of the standard solar spectrum.

The IPCE and APCE can be calculated from the formula[44,45]:

$$IPCE = \frac{J \times 1240}{\lambda \times P_{light}} \times 100\% \quad (12)$$

$$APCE = \frac{IPCE}{\eta_{abs.}(\lambda)} \quad (13)$$

$J$ refers to the photocurrent density (mA cm$^{-2}$) produced by the photoelectrode, whereas $P_{light}$ is the power density obtained at a specific wavelength ($\lambda$), respectively. $\eta_{abs.}$ is the efficiency of the light harvested, and it can be calculated from the obtained light absorbance curves:

$$\eta_{abs.} = (1 - 10^{-A}) \times 100\% \quad (14)$$

where the $A$ is light absorbance measured by a UV-vis spectroscopy.

**Computational details.** Density functional theory (DFT) calculations are performed with the Vienna ab initio simulation package (VASP)[46]. The Pedrew–Burke–Ernzerhof (PBE) functional is used to describe the electron exchange-correlation interactions[47], and the projected augmented approach wave is adopted to construct the pseudopotential for treat the ion-electron interaction[48]. A pane-wave cutoff energy of 400 eV and $4 \times 4 \times 1$ Γ-centered Monhorst-Pack k-point mesh are used for all calculations[49]. DFT-D3 method of Grimme is employed to describe the van der Waals interactions[50]. The dipole correction is taken into account for all calculations. The stoichiometric $4 \times 4$ one-layer 112-atom ZIS (001) surface with 64 S atoms, 16 Zn atoms, and 32 In atoms is built using the $ZnIn_2S_4$ unit cell[51] based on previous DFT calculations[52–54], because the hexagonal $ZnIn_2S_4$ has a sandwiched layered structure, and the layers are connected by van der Waals interaction. The interlayer interaction is rather weak that allows one to use a monolayer structure instead of a multi-layer model in the simulations[52]. Such settings not only maintain the major properties of the $ZnIn_2S_4$ because the catalytic reactions generally occur on the materials surface, but also reduce the computational cost. According to the previous DFT calculations on the $WO_{3-x}/ZnIn_2S_4$[53] and the $Li_2S_x/ZnIn_2S_4$ systems, we add an 8-atom ZnS cluster into the ZIS (001) surface and leads to the stoichiometric ZISZ system, Supplementary Fig. 7a, consistent with present experimental conditions. The ZISZ/Fe system is obtained by replacing an In atom or a Zn atom with an Fe atom on the topmost ZISZ surface, displayed in Supplementary Fig. 7b–e. To eliminate the interactions between periodic images, a vacuum region of 20 Å is added normal to the surface.

In order to verify the stability of the four geometries, we have performed the 8 ps adiabatic molecular dynamic simulations at 300 K in the microcanonical ensemble with a 1 fs time step after heating up all the systems to 300 K via repeated velocity rescaling lasting for 2 ps. The obtained evolution of total energy cumulative average over time for four systems oscillates only in a narrow window, shown in Supplementary Fig. 19, suggesting that the used structures are reasonable and thermodynamically stable. The four-electron OER processes are described as the aforementioned Eqs. (1)–(4).

The Gibbs free energy $G$ is calculated according to the equation as follows:

$$G = E + E_{ZPE} - TS - eU \quad (15)$$

where $E$, $E_{ZPE}$, and $S$ are the single point energy, zero-point energy, and entropy of the ZISZ and ZISZ/Fe slabs adsorbed with and without various oxygen intermediates, respectively. $U$ represents the potential versus standard hydrogen electrode. $T$ is set to 298.15 K.

The overpotential ($\eta$) for OER is estimated according to the following equation when $U$ equals to 0.

$$\eta = \frac{max\{\Delta G1, \Delta G2, \Delta G3, \Delta G4\}}{e} - 1.23 \quad (16)$$

Here, $\Delta G1$, $\Delta G2$, $\Delta G3$, and $\Delta G4$ denote the Gibbs free energy difference for each reaction, (1)–(4), respectively.

## Data availability

The data that support the findings of this study are available from the corresponding authors upon reasonable request, and the source data are provided with this paper (10.11922/sciencedb.00033).

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

## Acknowledgements

We acknowledge the support from the National Natural Science Foundation of China (52025028, 51972218), 1000 Youth Talents Plan, Key University Science Research Project of Jiangsu Province (17KJA430013), Six Talents Peak Project of Jiangsu Province, 333

High-level Talents Cultivation Project of Jiangsu Province, and the Priority Academic Program Development (PAPD) of Jiangsu Higher Education Institutions. We thank the support from the BL5.2 SUT-NANOTEC-SRLI XAS beamline, at the Synchrotron Light Research Institute, Nakhon Ratchasima, Thailand for XAS measurements.

## Author contributions

L.L. initially conceptualized and designed the project. L.M. and W.X. carried out all sample synthesis and characterization. L.L., L.M. and K.D. wrote the manuscript. R.L. and J.H. calculated the DFT results and wrote this part. P.K., X.Z. and Y.T. performed the XAS analyses. All authors contributed to the overall scientific interpretation and revised this paper.

## Competing interests

The authors declare no competing interests.
