## [Peer Review File · Nature Communications]

REVIEWER COMMENTS

Reviewer #1 (Remarks to the Author):

This manuscript tries to enhance the surface OER of ZnInS photoanode with ALD deposited Fe₂O₃, which is not novel enough for either the property or the mechanism. Besides, many conclusions are drawn arbitrarily and are confusing. Therefore, this work cannot be recommended for publication unless dressing the following questions.

1. The authors claim that the Fe-In-S atomic structure formed, which is also the absolute proof for DFT calculation. However, this result is questionable, especially with ALD flow for generating Fe₂O₃. Besides, the peak position of XANES spectra reveals the Fe atoms are more likely ascribed to be those in Fe₂O₃. The peaks in Figure 2f assigned to Fe-S and Fe-In are also questionable.
2. The models for DFT calculations in Figure S7 and S15 are unreasonable. The surface structural stability of the model in Figure S15 should be tested.
3. Fig. 3a shows the onset potential (the start potential of photocurrent) of ZISZ/Fe is positive than ZIS and ZISZ. Use the potential of $J=0.1\text{ mA cm}^{-2}$ as V_{on} is not convincing and the original OCP data are not provided. Besides, the decrease of V_{on} is not definitely determined by surface OER barriers. These articles can give a good ref. (J. Catal. 2013, 304, 86-91. Angew. Chem. Int. Ed. 2013, 52, 12692-12695)
4. The self-oxidation decomposition potential of ZnInS is more negative than the OER potential, and there are many possibilities to exhibit photoanodic current. For example, the photo-generated holes cause the self-decomposition of photoanodes. The authors haven't excluded the self-oxidation of photoanodes in 0.5 M Na₂SO₄.
5. The intensity-modulated photo-current spectroscopy (IMPS) spectroscopy related the charge transfer process, the high-frequency response corresponds to the charge transport and relaxation inside the photo-anode, whilst a semicircle in the low-frequency region is due to the competition between interfacial charge transfer and recombination. The IMPS results of Supplementary Figure 18., shows the frequency at the minimum imaginary part of ZISZ is higher than ZISZ/Fe, it means the electron transfer time of ZISZ/Fe is longer than ZISZ, it's converse to the conclusion of that the surface charge-transfer efficiency for ZISZ/Fe has a huge improvement compared with ZIS and ZISZ. More calculation details should be given.

Reviewer #2 (Remarks to the Author):

The work has reported an efficient ZnInS based photoanode with the modification of Fe-In-S group. It's encouraging to see that the Fe-In-S group has dramatically promoted the surface reaction performance of the ZnInS based photoelectrode. The manuscript is well written and should be able to attract the broad interest of readers in the relevant fields. For the authors to further improve the manuscript quality, I would suggest the authors to consider and clarify the following issues, as listed below:

- (1) The photoelectrodes exhibit high photocurrent density, so what's the key evidence to confirm the current is generated by water oxidation process? The first principle investigation of the surface OER process on the surface may not represent the case in the experimental research. Moreover, the stability investigation is also very important for claiming the efficiency of the prepared photoelectrode. Note that the photocorrosion can also produce high photocurrent, so the authors are encouraged to further check the key mechanism.
- (2) Lines 314-316, the ABPE calculation process should be calculated based on two-electrode system, where the voltage can be termed as the applied bias. More discussion can be referred to the following paper: Chen et al., Journal of Materials Research, 2010, 25(1): 3.
- (3) Lines 162-164, the authors claimed "the improved performance is attributed to the increased

light absorption.....". In some cases, the observed improved light harvesting does not guarantee the improved performance. To check about this, this authors are suggested to provide the incident photo-to-current efficiency (IPCE) and absorbed photon-to-current efficiency (APCE).

(4) Lines 123-124, the authors claimed that the sample was etched by Ar ions for XPS and UPS measurements. However, Ar ions etching process can dramatically change the surface states due to the strong reactive capability of the Ar ions. In this scenario, the detected in-depth chemical states and valance band may not be reliable, any comments?

(5) As for the Fe-In-S, the authors called it a "group". Is this more like a "cluster", but not a group with well-identified composition and stoichiometry. In addition, they authors should justify why the surface catalyst is Fe-In-S based materials, but not Fe-S based ones? The latter may be more likely for the ALD deposition process.

(6) In discussion the catalytic capability of Fe-In-S, it is suggested to carry out the LSV measurement without illumination.

Reviewer #1 (Remarks to the Author):

This manuscript tries to enhance the surface OER of ZnInS photoanode with ALD deposited Fe_2O_3 , which is not novel enough for either the property or the mechanism. Besides, many conclusions are drawn arbitrarily and are confusing. Therefore, this work cannot be recommended for publication unless dressing the following questions.

1. The authors claim that the Fe-In-S atomical structure formed, which is also the absolute proof for DFT calculation. However, this result is questionable, especially with ALD flow for generating Fe_2O_3 . Besides, the peak position of XANES spectra reveals the Fe atoms are more likely ascribed to be those in Fe_2O_3 . The peaks in Figure 2f assigned to Fe-S and Fe-In are also questionable.

[Author Reply]: Many thanks for your kind support and efforts in reviewing our manuscript. According to the data in this manuscript, there is no Fe_2O_3 on the surface of samples. Some detailed explanations and the further description of XANES spectra are provided:

1) The peak position of XANES spectra does not reveal the Fe atoms ascribed to be those in Fe_2O_3 . In the area enclosed by the red dotted line in Fig. R1a, unlike Fe_2O_3 standard sample, there is no characteristic signal at the energy range between 7140-7165 eV of our samples. Again, this can be used to confirm the difference in atomic structure around Fe ions in our samples compared with Fe_2O_3 standard sample.

2) The peaks in Fig. 2f assigned to Fe-S and Fe-In are reasonable. Due to the larger atomic radial distances of S and in comparison to those of O and Fe in Fe_2O_3 , thus, the peak positions

in R-space of Fe-S and Fe-In are located at the longer R distances (see the Fe K-edge r-space of Fe₂O₃ in Fig. R1b). These results provide strong evidences about the existence of Fe-In-S in our samples.

3) There is no Fe₂O₃ on the surface of the photoelectrode. Although the ALD deposition procedure involves ferrocene and ozone that used as iron and oxygen precursors, which are more likely to generate Fe₂O₃. Under normal condition, the ALD flow is used for generating Fe₂O₃, especially on various metal oxide materials. However, there are some difference in this manuscript when the substrate material is ZnInS. There is a new mechanism for this special case in our results. The relevant proof data are as follows:

Firstly, the X-ray absorption spectroscopy (XAS) of the Fe L₃ absorption edges (Fig. R1a-c) indicates there is no oxides formed. Fig. R1a and b shows the different characteristics of curves from standard FeO and Fe₂O₃ samples, manifesting that no oxides are formed. In addition, the peaks at 2.1 Å and 3.5 Å are assigned to Fe-S bond and Fe-In bond (Fig. R1c), respectively, and the content of Fe-S bond increases with the increased deposition cycles. These characterizations indicate that the Fe-In-S atomical structure is formed at the surface of photoanode and there is no Fe₂O₃.

Secondly, in order to confirm that Fe₂O₃ is not produced through the ALD deposition process based on ZnInS, the X-ray diffraction (XRD) pattern of ZISZ/Fe with 700 cycles is shown in Fig. R1d. The characteristic peaks of Fe₂O₃ do not appear even if the sample already contains a large amount of Fe element, indicating the existence of Fe is not in the form of Fe₂O₃ during the ALD deposition process.

Thirdly, to confirm that there exists no Fe₂O₃ on the surface of photoanode, the XPS of

samples treated under different ALD cycles is shown in Supplementary Fig. 6. With the increased cycles, the peak intensity of In gradually decreases while that of Fe increases. The reduced intensity of In results from the replacement of Fe to build Fe-In-S group, indicating there is no Fe₂O₃ generation.

Finally, we also measured the XPS at different depths through Ar ion etching (Supplementary Fig. 8). As the etching depth increases, the peak intensity of Zn gradually increases and its peak position shifts toward higher energy (Supplementary Fig. 8a). The peak intensity of In also increases while the peak position shifts toward lower energy (Supplementary Fig. 8b). When the etching depth increases to 50 nm, the signal of Fe substantially disappears (Supplementary Fig. 8c), but the peak position of O shifts toward higher energy (Supplementary Fig. 8d). According to these data, we can get the following conclusions: 1) Fe is mainly deposited on the surface with a thickness of about 50 nm. 2) The shift of Zn peak toward higher energy and O peak toward lower energy are due to the formed Zn-O. The Zn-O bond is shorter than Zn-S bond, thus the binding energy of Zn-O is higher. 3) The opposite trend of the peak intensity of In and Fe also indicates the successful substitution. In a word, the lattice O appears inside the samples and Fe exists on the surface, respectively. Thus it is impossible that Fe₂O₃ can be formed on the surface of the samples.

In summary, through the above characterizations, we conclude that the Fe₂O₃ can not form during the ALD flow and there is no Fe₂O₃ in the samples. Unlike the normal ALD deposition process, there is a new mechanism for the present material system.

Fig. R1. Characterizations of photoanodes. **a** The Fe K-edge XANES spectra. **b** The FT-EXAFS of Fe₂O₃ and ZISZ/Fe. **c** The FT-EXAFS of ZISZ/Fe with different ALD cycles (40, 70, and 100 cycles); k = wave vector and $\chi(k)$ = oscillation as a function of the photoelectron wavenumber. **d** XRD pattern of ZISZ/Fe with 700 cycles.

Supplementary Figure 6. XPS spectra for In 3d and Fe 2p. XPS spectra of (a) In 3d and (b)

Fe 2p of ZISZ with different ALD cycles.

Supplementary Figure 8. XPS spectra to illustrate the states of photoanodes. XPS spectra from ZISZ/Fe at different etching depth. (a) Zn 2p, (b) In 3d, (c) Fe 2p, and (d) O 1s.

The peak region (7110–7170 eV) suggests that the photoanode contains different Fe content.

Unlike Fe₂O₃ standard sample, there is no feature at the energy range between 7140-7165 eV of our prepared samples. The different characteristics of curves from standard FeO and Fe₂O₃ samples manifests that no oxides are formed.

Fig. 2. Characterizations of photoanodes. a Zn 2p, b In 3d, c S 2p, and (d) Fe 2p XPS of ZIS, ZISZ, and ZISZ/Fe. e The Fe K-edge XANES spectra. f The Fe K-edge EXAFS as functions $k^2\chi(k)$ and FT-EXAFS of ZISZ/Fe with different ALD cycles (40, 70, and 100 cycles); k = wave vector and $\chi(k)$ = oscillation as a function of the photoelectron wavenumber.

2. The models for DFT calculations in Figure S7 and S15 are unreasonable. The surface structural stability of the model in Figure S15 should be tested.

[Author Reply]: According to your suggestion, we have added the equivalent circuit in the manuscript.

We thank the Reviewer for raising this issue. The ZnIn_2S_4 (ZIS) surface is represented by a ZIS (001) slab in our simulations, which was created using the ZnIn_2S_4 unit cell¹ widely used by previous DFT calculations². The introduction of an 8-atom ZnS cluster on the ZIS (001) surface forms the ZISZ system, referring to the $\text{WO}_{3-x}/\text{ZnIn}_2\text{S}_4$ system³. The replacement of an In atom or a Zn atom with an Fe atom in the ZISZ system leads to the ZISZ/Fe system. The substitution is energetically favorable because the calculated formation energy is negative. To further examine the stability of four systems, we carried out adiabatic molecular dynamic simulations at room temperature lasting for 8 picoseconds and plotted the evolution of total energy of each system in Supplementary Fig. 19. The small energy fluctuations around the equilibrium position demonstrate that all geometries used in the present work are reasonable and thermodynamically stable. We have replotted Supplementary Fig. 19 and combined it with Supplementary Fig. 7 during the revision.

We have added the corresponding discussion in the computational details and cited the relevant literatures in the revised manuscript.

1. Hu, X., Yu, J., Gong, J., Li, Q., Rapid mass production of hierarchically porous ZnIn_2S_4 submicrospheres via a microwave-solvothermal process. *Cryst. Growth Des.* **7**, 2444-2448 (2007).
2. Shi, X., Mao, L., Yang, P., Zheng, H., Fujitsuka, M., Zhang, J., Majima, T., Ultrathin

ZnIn₂S₄ nanosheets with active (110) facet exposure and efficient charge separation for cocatalyst free photocatalytic hydrogen evolution. *Appl. Catal. B Environ.* **265**, 118616 (2020).

3. Luo, D., et al. Highly efficient photocatalytic water splitting utilizing a WO_{3-x}/ZnIn₂S₄ ultrathin nanosheet Zscheme catalyst. *J. Mater. Chem. A* **9**, 908-914 (2021).

The 112-atom ZIS (001) surface is built using the ZnIn₂S₄ unit cell⁵¹, which is investigated by previous DFT calculations⁵². According to the calculations performed by Luo et al. on the WO_{3-x}/ZnIn₂S₄ system⁵³, we introduce an interface of an 8-atom ZnS cluster into the ZIS(001) surface and thus forms the ZISZ system, Supplementary Fig. 7a. The ZISZ/Fe system is obtained by replacing an In atom or a Zn atom with an Fe atom on the topmost ZISZ surface, displayed in Supplementary Fig. 7b-d. To eliminate the interactions between periodic images, a vacuum region of 20 Å is added normal to the surface.

In order to verify the stability of the four geometries, we have performed the 8 picoseconds adiabatic molecular dynamic simulations at 300 K in the microcanonical ensemble with a 1 fs time step after heating up all the systems to 300 K via repeated velocity rescaling lasting for 2 picoseconds. The obtained evolution of total energy for four systems oscillates only in a narrow window, showed in Supplementary Fig. 19, suggesting that the used structures are reasonable and thermodynamically stable. The four-electron OER processes are described as the aforementioned equations (1)–(4).

51. Hu, X., Yu, J., Gong, J., Li, Q., Rapid mass production of hierarchically porous ZnIn₂S₄

submicrospheres via a microwave-solvothermal process. *Cryst. Growth Des.* **7**, 2444-2448 (2007).

52. Shi, X., Mao, L., Yang, P., Zheng, H., Fujitsuka, M., Zhang, J., Majima, T., Ultrathin ZnIn₂S₄ nanosheets with active (110) facet exposure and efficient charge separation for cocatalyst free photocatalytic hydrogen evolution. *Appl. Catal. B Environ.* **265**, 118616 (2020).

53. Luo, D., et al. Highly efficient photocatalytic water splitting utilizing a WO_{3-x}/ZnIn₂S₄ ultrathin nanosheet Zscheme catalyst. *J. Mater. Chem. A* **9**, 908-914 (2021).

Supplementary Figure 7. Models for formation energy calculations. Simulation cell showing the optimized geometries of (a) pristine ZISZ (001) surface and ZISZ/Fe structures of Fe substituting to (b, c) Zn and (d) In. The formation energies (E_f) are listed below each geometry (b-d) and are obtained using the formula: $E_f = E_{\text{ZISZ/Fe}} - E_{\text{ZISZ}} - E_{\text{Fe}} + E_{\text{Zn(In)}}$. Here, $E_{\text{ZISZ/Fe}}$ and E_{ZISZ} are the total energy of the ZISZ/Fe and ZISZ systems, and E_{Fe} and $E_{\text{Zn(In)}}$ correspond to the energy of a single Fe, and Zn or In atom, respectively, calculated using the unit cell of each bulk material. Greater negative value of formation energy means easier substitution. Correspondingly, the estimation of OER efficiencies between pristine with and without Fe doping are calculated and compared between the systems (a) and (d).

Supplementary Figure 19. Evolution of total energy for four systems oscillates only in a narrow window. Evolution of total energy of the 8 ps adiabatic MD trajectories in the microcanonical ensemble of pristine ZISZ (a) and (b-d) ZISZ/Fe systems. Small energy fluctuations indicate that the four systems used in the present work are thermodynamically stable.

3. Fig. 3a shows the onset potential (the start potential of photocurrent) of ZISZ/Fe is positive than ZIS and ZISZ. Use the potential of $J=0.1\text{mA cm}^{-2}$ as V_{on} is not convincing and the original OCP data are not provided. Besides, the decrease of V_{on} is not definitely determined by surface OER barriers. These articles can give a good ref. (J. Catal. 2013, 304, 86-91. Angew. Chem. Int. Ed. 2013, 52, 12692-12695).

[Author Reply]: Many thanks for your suggestions in reviewing our manuscript. Both the articles you provided explain the V_{on} in detail. According to these articles, we have made some

changes and provided the original OCP data in this manuscript.

Refer to these works, we have redefined the potential of $J=0.02 \text{ mA cm}^{-2}$ as V_{on} (defined as the potential at which 0.02 mA cm^{-2} was first measured). In addition, the influencing factors of V_{on} are not only the surface OER barriers, but also the photovoltage. The greater photovoltage and reduced kinetic overpotential can cause the cathodic shift of V_{on} in photoanode. Thus, in order to better explain the change of V_{on} , we added the new explanation about the decrease of V_{on} in the manuscript according to these articles.

the V_{on} is related to photovoltage and surface OER barriers, and the photovoltage can be determined by the open-circuit potential (OCP) measurement. The greater photovoltage and reduced surface OER barriers can cause the cathodic shift of V_{on} in photoanode. The V_{on} of ZISZ has a slightly positive shift compared with ZIS, as is also revealed by OCP (Supplementary Fig. 11), that ZISZ possesses the smallest value. Although the OCP of ZISZ/Fe is still smaller than ZIS, it possesses a cathodic V_{on} . This may be due to the fact that the existence of Fe-In-S group reduces the surface electrochemical reaction barrier of ZISZ/Fe photoanode^{41,42}.

41. Yang, X., Du, C. Liu, R. Xie, J., Wang, D. Balancing photovoltage generation and charge-transfer enhancement for catalyst-decorated photoelectrochemical water splitting: A case study of the hematite/ MnO_x combination. *J. Catal.* **304**, 86-91 (2013).

42. Du, C., et al. Hematite-based water splitting with low turn-on voltages. *Angew. Chem. Int. Ed.* **125**, 12924-12927 (2013).

Supplementary Figure 11. The photovoltage positions. (a) OCP measurements under dark and light conditions and (b) OCP values of ZIS, ZISZ, and ZISZ/Fe photoanodes, respectively.

4. The self-oxidation decomposition potential of ZnInS is more negative than the OER potential, and there are many possibilities to exhibit photoanodic current. For example, the photo-generated holes cause the self-decomposition of photoanodes. The authors haven't excluded the self-oxidation of photoanodes in 0.5 M Na₂SO₄.

[Author Reply]: In this manuscript, we also excluded the photoanode in Na₂S/Na₂SO₃. However, this electrolyte is used as hole scavengers, therefore we did not use it as the main test system. The reasons why we select 0.5 M Na₂SO₄ as electrolyte are as follows:

1) The main innovation in this paper is the construction of cocatalysts, promoting the rapid transfer of photogenerated holes. If other electrolytes are used for measuring the photoanode, such as 0.5 M Na₂SO₃ or Na₂S/Na₂SO₃, etc., that is not conducive to this experiment to characterize the role of cocatalysts since these electrolytes are usually used as hole scavengers.

2) For most of articles, 0.5 M Na₂SO₃ or Na₂S/Na₂SO₃ are used as hole scavengers to obtain

the injection efficiency, proving the effectiveness of cocatalysts. Under normal conditions, it is not recommended to directly apply this electrolyte as a test system.

3) Regarding the self-oxidation process you mentioned, if the process exists in this electrolyte, the dark current should have an obvious oxidation peak, but the dark current is obviously normal (Fig. R2). In addition, many previous results were also based on this electrolyte for photoelectrochemical (PEC) testing as shown in Table R1.

Fig. R2. the LSV measurement without illumination of ZIS, ZISZ, and ZISZ/Fe.

Table R1. The electrolyte used by Na₂SO₄ in sulfide based photoanodes.

Photoelectrode materials	Electrolyte	References
V _s -CdIn ₂ S ₄	Na ₂ SO ₄	1
In ₂ S ₃ /Bi ₂ S ₃	Na ₂ SO ₄	2
WO ₃ /In ₂ S ₃	Na ₂ SO ₄	3
In ₂ S ₃ /In ₂ O _{3-x}	Na ₂ SO ₄	4
ZnO/ZnS	Na ₂ SO ₄	5
CdS Nanorod/ hexagonal SnS ₂	Na ₂ SO ₄	6
ZnIn ₂ S ₄ /RGO/ZnO	Na ₂ SO ₄	7
Vertical SnS _x /CdS	Na ₂ SO ₄	8
SnS ₂ /SnS/OS	Na ₂ SO ₄	9
WO ₃ /ZnIn ₂ S ₄	Na ₂ SO ₄	10
SnS ₂ /N-SnS ₂	Na ₂ SO ₄	11
ZnInS/Fe-In-S	Na ₂ SO ₄	This work

1. Wang, H., et al. Highly active deficient ternary sulfide photoanode for photoelectrochemical water splitting. *Nat. Commun.* **11**, 3078 (2020).
2. Xiong, Y., et al. Highly efficient photoelectrochemical water oxidation enabled by enhanced interfacial interaction in 2D/1D In₂S₃@Bi₂S₃ heterostructures. *J. Mater. Chem. A* **8**, 5612-5621 (2020).
3. Tian, W., Chen, C., Meng, L., Xu, W., Cao, F., Li, L. PVP treatment induced gradient oxygen

- doping in In₂S₃ nanosheet to boost solar water oxidation of WO₃ nanoarray photoanode. *Adv. Energy Mater.* **10**, 1903951 (2020).
4. Hou, J., et al. Atomically thin mesoporous In₂O_{3-x}/In₂S₃ lateral heterostructures enabling robust broadband-light photo-electrochemical water splitting. *Adv. Energy Mater.* **8**, 1701114 (2017).
 5. Hassan, M. A., Waseem, A., Johar, M. A., Bagal, I. V., Ha, J.-S., Ryu, S.-W. Single-step fabrication of 3D hierarchical ZnO/ZnS heterojunction branched nanowires by MOCVD for enhanced photoelectrochemical water splitting. *J. Mater. Chem. A* **8**, 8300-8312 (2020).
 6. Fu, Y., et al. Phase-modulated band alignment in CdS nanorod/SnS_x nanosheet hierarchical heterojunctions toward efficient water splitting. *Adv. Funct. Mater.* **28**, 1706785 (2018).
 7. Bai, Z., Yan, X., Kang, Z., Hu, Y., Zhang, X., Zhang, Y. Photoelectrochemical performance enhancement of ZnO photoanodes from ZnIn₂S₄ nanosheets coating. *Nano Energy* **14**, 392-400 (2015).
 8. Giri, B., et al. Balancing light absorption and charge transport in vertical SnS₂ nanoflake photoanodes with stepped layers and large intrinsic mobility. *Adv. Energy Mater.* **9**, 1901236 (2019).
 9. Meng, L., et al. A plasma-triggered O-S bond and P-N junction near the surface of a SnS₂ nanosheet array to enable efficient solar water oxidation. *Angew. Chem. Int. Ed.* **58**, 16668-16675 (2019).
 10. Shi, X., Dong, F., Dai, C., Ye, X., Yang, P., Zheng, L., Zheng, H. WO₃/ZnIn₂S₄ heterojunction photoanodes grafting silane molecule for efficient photoelectrochemical water splitting. *Electrochim. Acta.* **361**, 137017 (2020).

11. Wu, Y., Liu, X., Zhang, H. Li, J., Zhou, M., Li, L., Wang, Y. Atomic sandwiched p-n homojunctions. *Angew. Chem. Int. Ed.* **59**, 1-7 (2020).

5. The intensity-modulated photo-current spectroscopy (IMPS) spectroscopy related the charge transfer process, the high-frequency response corresponds to the charge transport and relaxation inside the photo-anode, whilst a semicircle in the low-frequency region is due to the competition between interfacial charge transfer and recombination. The IMPS results of Supplementary Figure 18., shows the frequency at the minimum imaginary part of ZISZ is higher than ZISZ/Fe, it means the electron transfer time of ZISZ/Fe is longer than ZISZ, it's converse to the conclusion of that the surface charge-transfer efficiency for ZISZ/Fe has a huge improvement compared with ZIS and ZISZ. More calculation details should be given.

[Author Reply]: According to your suggestion, we have added more calculation details in this manuscript.

In this manuscript, IMPS was performed to study the charge transfer and transit dynamics processes, including surface charge-transfer efficiency (η_{tran}), surface charge-transfer rate constants (K_{tran}), and surface charge recombination rate constants (K_{rec}). The relationship between them can be described by the following formula:

$$\eta_{\text{tran}} = K_{\text{tran}} / (K_{\text{tran}} + K_{\text{rec}})$$

$$K_{\text{tran}} + K_{\text{rec}} = 2\pi\nu$$

the calculation details as shown in Fig. R3, thus through two formula we can obtain the η_{tran} , K_{tran} , and K_{rec} .

On the other hand, as you mentioned, the frequency at the minimum imaginary part of ZISZ is higher than ZISZ/Fe, which means the electron transfer time of ZISZ/Fe is longer than ZISZ.

This result can be interpreted by the following formula:

$$\tau_d = \frac{1}{2 \times 3.14 \times f_{min}}$$

τ_d is electron transfer time, f_{min} is the frequency of the lowest point of the imaginary part in IMPS, which is shown in Fig. R3.

The longer τ_d is due to the less interface recombination. The increased charge-transfer rate and the reduced charge recombination rate of holes cause the reduced interface recombination. Thus the longer τ_d of ZISZ/Fe is not converse to the conclusion of that the surface charge-transfer efficiency for ZISZ/Fe has a huge improvement compared with ZIS and ZISZ. In the part of photoanode, the surface OER process is mainly dominated by photogenerated holes. The extended τ_d does not explain the dull OER process on the surface of photoanode. Owing to the change of the band energy and the reduced charge recombination rate of holes, the τ_d should be longer in ZISZ/Fe, which is consistent with the reduced bulk recombination and the results of PL and TRPL.

In addition, electron transfer time (τ_d) of photoanodes can be calculated form the formula:

$$\tau_d = \frac{1}{2 \times 3.14 \times f_{min}}$$

where the f_{min} is the frequency of the lowest point of the imaginary part in IMPS. The IMPS results shows longer τ_d of ZISZ/Fe owing to the frequency at the minimum imaginary part of ZISZ is higher than ZISZ/Fe (Supplementary Figure 18), which is consistent with the reduced bulk recombination.

Fig. R3 The IMPS results and the detailed Calculation equation.

Reviewer #2 (Remarks to the Author):

The work has reported an efficient ZnInS based photoanode with the modification of Fe-In-S group. It's encouraging to see that the Fe-In-S group has dramatically promoted the surface reaction performance of the ZnInS based photoelectrode. The manuscript is well written and should be able to attract the broad interest of readers in the relevant fields. For the authors to further improve the manuscript quality, I would suggest the authors to consider and clarify the following issues, as listed below:

(1) The photoelectrodes exhibit high photocurrent density, so what's the key evidence to confirm the current is generated by water oxidation process? The first principle investigation of the surface OER process on the surface may not represent the case in the experimental research. Moreover, the stability investigation is also very important for claiming the efficiency of the prepared photoelectrode. Note that the photocorrosion can also produce high photocurrent, so the authors are encouraged to further check the key mechanism.

[Author Reply]: According to your suggestion, some further explanation has been added.

If the oxidation process occurs during the material rather than the water, this process should still exist without illumination. Since the theoretical water splitting voltage is above 1.23 V, the advantage of PEC water splitting is that solar energy can provide additional energy to achieve photoelectricity complementation. Therefore, water could be splitted under a low voltage (less than 1.23 V). Unless the measured current value is not from the oxidation of water, but from the oxidation reaction of the material itself, in the absence of light for supplementation, it is theoretically impossible to split water under low voltages, and there will be no large current

value. Similarly, for the photoanode, if the photocurrent generated in PEC water splitting does not come from the oxidation process of water, then there should still be a higher dark current or oxidation peak in the dark state. According to the discussions above, if the current value obtained does not come from the oxidation process of water, a relatively high dark current or redox peak will appear in J - V curves without illumination. However, in this work, the LSV measurement without illumination in Fig. R4a does not show the abnormal curve, and the oxidation peak does not appear. Thus, the photoelectrode with a high J is generated by water oxidation process under illumination rather than the self-oxidation of photoanodes.

In addition, we conducted a stability test in Fig. R4b, which shows the stability has been improved to certain extent after loading Fe-In-S cocatalyst, but there are still some gaps compared with some metal oxides. The explanation about the stability of ZnInS probably has the following points:

- 1) Metal sulfides are inherently unstable. During the test, the material acts as photoanode and generates a large number of photo-generated electron-hole pairs under illumination. Owing to its poor surface OER reaction kinetics of photoanode, a large number of photo-generated holes accumulate on the surface of the photoanode, causing serious self-corrosion. According to IMPS data, we can find that the interface transfer constant has been greatly improved after loading Fe-In-S, which is the reason why the stability of ZISZ/Fe has been improved.

- 2) Another important reason about the stability of ZnInS is that the elements are unstable in the test process. We carried out the ICP test (Table R2) on the tested electrolyte and found that it contains a lot of Zn, In, and S elements, especially Zn and S elements, which shows that the photoanode instability is due to the dissolution of its own elements in the process of stability

testing. Currently, there is no good solution for the stability of ZnInS and this is not the core of this article. Some works focus on the stability design such as adding some metal ions to the electrolyte to inhibit element dissolution, or loading a barrier layer of metal oxide. However, these schemes are all ready-made and can not reflect the innovation of this work. Thus, the stability design part is only to verify the Fe-In-S clusters to promote the transfer of photogenerated holes by detecting the improvement of stability.

Fig. R4. PEC performance characterizations. a Linear sweep voltammogram curves. **b** I-T curves of ZIS, ZISZ, and ZISZ/Fe.

Table R2. the elements content of Zn, In, and S in the electrolyte after the I-T measures.

Elements		Content (mg/L)
	Zn	3.9576
Na ₂ SO ₄	In	0.1711
	S	14.8812

(2) Lines 314-316, the ABPE calculation process should be calculated based on two-electrode system, where the voltage can be termed as the applied bias. More discussion can be referred to the following paper: Chen et al., Journal of Materials Research, 2010, 25(1): 3.

[Author Reply]: Many thanks for your kind support and efforts in reviewing our manuscript.

According to this article, ABPE does come from a two-electrode system, but according to

the literature on PEC published in recent years, the ABPE should be measured via a three-electrode system. The advantages of this are:

1) The advantage of PEC lies in the complementarity of optoelectronics. The decomposition voltage of water is 1.23 V vs. RHE. Therefore, when comparing the performance of PEC, the comparison is often performed at 1.23 V vs. RHE to highlight the advantages of optoelectronics. However, the pH provided by different electrolytes is inconsistent. For unified comparison, we need to use a three-electrode system for unified comparison. The comparison of ABPE should be standardized. As a relatively uniform variable, using the three-electrode system to RHE to unify the standard to RHE is conducive to direct comparison between different electrolytes. And in the three-electrode system, the bias voltage provided by the system also exists as an external bias voltage.

2) The ABPE calculation process via a three-electrode system is also utilized by some previous articles:

1. Ye, K. H., et al. Enhancing photoelectrochemical water splitting by combining work function tuning and heterojunction engineering. *Nat. Commun.* **10**, 3687 (2019).
2. Wang, S., et al. In situ formation of oxygen vacancies achieving near - complete charge separation in planar BiVO₄ photoanodes. *Adv. Mater.* **32**, 2001385 (2020).
3. Su, J., Hisatomi, T., Minegishi, T., Domen, K. Wang, Y. Enhanced photoelectrochemical water oxidation from CdTe photoanodes annealed with CdCl₂. *Angew. Chem. Int. Ed.* **132**, 2-9 (2020).
4. Zhang, H., et al. Gradient tantalum-doped hematite homojunction photoanode improves both photocurrents and turn-on voltage for solar water splitting. *Nat. Commun.* **11**, 4622

(2020).

5. Pan, J., et al. Activity and stability boosting of oxygen-vacancy-rich BiVO₄ photoanode by NiFe-MOFs thin layer for water oxidation. *Angew. Chem. Int. Ed.* **133**, 1453-1460 (2021).

(3) Lines 162-164, the authors claimed “the improved performance is attributed to the increased light absorption.....”. In some cases, the observed improved light harvesting does not guarantee the improved performance. To check about this, this authors are suggested to provide the incident photo-to-current efficiency (IPCE) and absorbed photon-to-current efficiency (APCE).

[Author Reply]: According to your suggestion, we have added the IPCE and APCE in the manuscript.

The IPCE and APCE can be calculated from the formula^{44,45}:

$$\text{IPCE} = \frac{J \times 1240}{\lambda \times P_{\text{light}}} \times 100\%$$

$$\text{APCE} = \frac{\text{IPCE}}{\eta_{\text{abs.}}(\lambda)}$$

J refers to the photocurrent density (mA cm⁻²) produced by the photoelectrode, whereas

P_{light} is the power density obtained at a specific wavelength (λ), respectively.

$\eta_{\text{abs.}}$ is the efficiency of the light harvested, and it can be calculated from the obtained light absorbance curves:

$$\eta_{\text{abs.}} = (1 - 10^{-A}) \times 100\%$$

where the A is light absorbance measured by a UV-vis spectroscopy.

44. Chen, Z., et al. Accelerating materials development for photoelectrochemical hydrogen production: Standards for methods, definitions, and reporting protocols. *J. Mater. Res.*, **25**, 3-16 (2010).

45. Fu, Y., et al. Surface electronic structure reconfiguration of hematite nanorods for efficient photoanodic water oxidation. *Sol. RRL.*, **4**, 1900349 (2020).

The capability of light harvesting gradually increases with the decreased bandgap for ZIS, ZISZ, and ZISZ/Fe (Fig. 3b). The incident photo-to-current efficiency (IPCE) and absorbed photon-to-current efficiency (APCE) show the increased IPCE and APCE value over the whole wavelength range (Supplementary Fig. 13), indicating the outstanding PEC performance of ZISZ/Fe. The IPCE and APCE is in accordance with the variation of photocurrents shown in Fig. 3a. This superior performance confirms the excellent light absorption, η_{sep} , and injection efficiency (η_{inj}), which we will discuss as follows.

However, theoretical photocurrent density ($J_{abs.}$) is a unified standard for judging the light absorption of the ZIS, ZISZ, and ZISZ/Fe photoanodes. The unity converted photocurrent density ($J_{abs.}$) is calculated by the following formula:

$$J_{abs.} = \int_{300}^{\lambda_{max}} \frac{\lambda \times \eta_{abs.}(\lambda) \times E(\lambda)}{1240} d(\lambda)$$

where λ_{max} is the maximum light absorption edge of a photoelectrode, λ (nm) is the light wavelength, $E(\lambda)$ is the power density (mW cm^{-2}) at a specific wavelength (λ) of the standard

solar spectrum., and $\eta_{\text{abs.}}$ is the efficiency of the light harvested, and it can be calculated from the obtained light absorbance curves:

$$\eta_{\text{abs.}} = (1 - 10^{-A}) \times 100\%$$

where the A is light absorbance measured by a UV-vis spectroscopy.

According to these formula, we can calculate the theoretical $J_{\text{abs.}}$ of the three electrodes, and the value is 4.47, 6.07 and 6.54 mA cm⁻², respectively.

In summary, we claimed “the improved performance is attributed to the increased light absorption, η_{sep} , and injection efficiency (η_{inj}).”

Supplementary Figure S13. (a) IPCE curves and (b) APCE values of ZIS, ZISZ, and ZISZ/Fe photoanodes, respectively.

(4) Lines 123-124, the authors claimed that the sample was etched by Ar ions for XPS and UPS measurements. However, Ar ions etching process can dramatically change the surface states due to the strong reactive capability of the Ar ions. In this scenario, the detected in-depth chemical states and valance band may not be reliable, any comments?

[Author Reply]: Thanks for your suggestion. The Ar ions etching process can dramatically change surface states due to the strong reactive capability of Ar ions. However, in this manuscript, the affect is negligible and the detected in-depth chemical states and valance band is reliable. The reasons as follows:

1) The Ar ions are relatively stable in a vacuum environment, and are frequently used as etching materials. Under normal circumstances, Argon ions are accelerated by the anode electric field to physically bombard the surface of the sample to achieve the effect of etching, which belongs to pure physical etching.

2) In the XPS measurements, the detection depth is less than 10 nm. In order to detect the composition of surface deeper than 10 nm, the Ar ions etching process is used to peel off embedded materials generally. In other words, since most of the samples are in the atmospheric environment before XPS analysis, it is easy to cause adsorption pollution. For measuring the true information of sample surface, the sample surface must be pre-cleaned. The commonly used surface cleaning technology is argon ion etching technology.

3) If Ar ions etching changes the surface states due to the reductive nature of Ar, the chemical states will be reduced. However, according the Supplementary Figure 8, the Zn and O is changed owing to the oxidation. Thus, we think the impact of Ar ions etching is negligible.

4) The Ar ions etching process is used widely, especially at XPS and UPS measurements. For

example, some references can illustrate this:

1. Yu, Y., Huang Y., Yu, Y., Shi, Y., Zhang, B. Design of continuous built-in band bending in self-supported CdS nanorodbased hierarchical architecture for efficient photoelectrochemical hydrogen production. *Nano Energy*, **43**, 236-243 (2020).
2. Luo, Z., Li, C., Liu, S., Wang, T., Gong, J., Gradient doping of phosphorus in Fe₂O₃ nanoarray photoanodes for enhanced charge separation. *Chem. Sci.*, **8**, 91-100 (2017).
3. Kong, D., et al., Ti-gradient doping to stabilize layered surface structure for high performance high-Ni oxide cathode of Li-Ion battery. *Adv. Energy Mater.*, **9**, 1901756 (2019).
4. Zhao, Y., et al., surface structural transition induced by gradient polyanion-doping in Li-rich layered oxides: implications for enhanced electrochemical performance. *Adv. Funct. Mater.*, **26**, 4760-4767 (2016).
5. Huang, H., et al., Oriented built-in electric field introduced by surface gradient diffusion doping for enhanced photocatalytic H₂ evolution in CdS Nanorods. *Nano Lett.*, **17**, 3803-3808 (2017).

Supplementary Figure 8. XPS spectra to illustrate the states of photoanodes. XPS spectra from ZISZ/Fe at different etching depth. (a) Zn 2p, (b) In 3d, (c) Fe 2p, and (d) O 1s.

(5) As for the Fe-In-S, the authors called it a “group”. Is this more like a “cluster”, but not a group with well-identified composition and stoichiometry. In addition, they authors should justify why the surface catalyst is Fe-In-S based materials, but not Fe-S based ones? The latter may be more likely for the ALD deposition process.

[Author Reply]: According to your suggestion, we have called the Fe-In-S as a “cluster”.

In addition, we justify the surface catalyst is Fe-In-S based materials, the reason as follows:

1) As you considered, the ALD deposition process is more likely for growing FeS, but the surface catalyst is Fe-In-S based materials rather than FeS. Firstly, the process of replacing In with Fe occurs on the surface. Secondly, we did not supplement the S source, so this deposition process is not an experiment for preparing FeS. Furthermore, when the replacement reaction of Fe occurs, the Fe-In-S exists as a whole in this area, it is not possible to isolate Fe-S alone. Therefore, we believe that Fe-In-S has a good catalytic effect on the surface.

2) According to the XPS data, we can know that when Fe enters the ZIS material, the reaction of Fe replaces In occurs. Furthermore, according to the DFT theoretical calculation model, we find that substitution of In with Fe requires smaller formation energy than replacement of Zn with Fe, which suggests that the former substitution is energetically favorable. Furthermore, the replaced Fe atom is distant from the Zn atom and facilitates to form Fe-In-S bond between the Fe dopant and its surrounding In and S atoms. Therefore, the Fe-In-S formed in this area as a whole catalyzes the oxidation of water.

3) The most important evidence is that we have performed XAS fine spectrum analysis on the sample and found that it is not consistent with the standard spectrum of FeS (Fig. R5). It proves that there is no FeS on the surface, therefore, the surface catalyst is Fe-In-S based materials. Furthermore, according to the Fe K-edge extended X-ray absorption fine structure (EXAFS) as functions $k^2\chi(k)$ and its Fourier-transformation (FT-EXAFS) ($FT-\kappa^2\chi(\kappa)$) for ZISZ/Fe (Fig. 2f), we can find that Fe-S and Fe-In appear on the surface of the sample, so we think that it cannot be simply named after FeS alone.

Fig. R5 The Fe K-edge XANES spectra of ZISZ/Fe with different ALD cycles (40, 70, and 100 cycles)

(6) In discussion the catalytic capability of Fe-In-S, it is suggested to carry out the LSV measurement without illumination.

[Author Reply]: Thanks for your suggestion We have provided the LSV measurement without illumination in the manuscript.

According to the dark current provided in this manuscript, we can find that Fe-In-S cluster exists as a good photo-cocatalyst. According to the J - V curves, the Fe-In-S cluster does not generate a significant current in the dark state (Fig. R6), but provides a significant performance improvement in the light state (Fig. 3a).

We know that electrocatalysts can reduce the water oxidation reaction barrier in the process of water splitting, but not all electrocatalysts can be used to build cocatalysts in PEC water splitting. It is necessary to ensure that the dark current is constant for constructing the cocatalysts, so that the impact of electrocatalytic water splitting can be avoided.

Fig. R6 the LSV measurement without illumination of ZIS, ZISZ, and ZISZ/Fe.

REVIEWER COMMENTS

Reviewer #1 (Remarks to the Author):

The authors have answered most of the questions that I concern. Therefore, I consider the manuscript in this version can be accepted.

Reviewer #2 (Remarks to the Author):

The revised manuscript has carefully considered all the comments of the reviewers and made further improvement. I would suggest the acceptance and only have one minor comment on the the IPCE and APCE data of the ZISZ/Fe photoanodes in the supplementary Figure S13, why the data show some drop in the wavelength range of 360 nm.

Reviewer #3 (Remarks to the Author):

This authors have synthesized ordered nanosheet arrays ZISZ/Fe photoanode material by ALD, which shows reasonably high OER activity. DFT calculation was used to study the OER mechanism to understand and verify the experimental results. However, I find some arguments are rather vague, and conclusions are not fully supported their results, and suggest that the authors carefully consider the following issues:

1. Hexagonal ZnIn₂S₄ is a layered material with the thickness of each layer about 10Å, and the layers are connected by van der Waals interaction. When constructing the ZIS model for DFT calculation, why is the one-layer model used instead of the multi-layer model?
2. The DFT model of ZISZ is stoichiometric, but is not without question. This model should be validated so as to be representative of the experimentally synthesized materials.
3. In the DFT model of ZISZ/Fe, how to determine the position of the replaced In atom when replacing In with Fe, and whether is the replacement of In in the middle layer considered? The authors have mentioned that the experimentally synthesized ZISZ/Fe has Zn-O bonds at the bottom, but the role of O is not clearly explained. Also, have the authors considered the influence of Zn-O bond in the DFT calculation?
4. In these DFT models, vacuum is added to eliminate the influence of periodic boundary conditions, but for these asymmetric models, is the dipole correction considered in the calculation? In addition, what are the oxidation state and spin state of Fe introduced in the system?
5. The entire structure of the OER four-step reaction calculated by DFT should be given in detail in the supplementary information, rather than just some partial structures in Figure 3e and Figure 3f. Moreover, the original total energy given in Supplementary Figure 19 can be converted into a cumulative average over time, will it be clearer?
6. Figure 2d and Supplementary Figure 6b analyze Fe 2p through XPS spectroscopy. As the number of cycles increases, the peak position and chemical composition of Fe have changed. How do you explain the change in chemical composition here?
7. Figure 2f is obtained by Fourier transform of the Fe k-edge EXAFS. As the number of ALD cycles increases, no obvious peak is seen at the position corresponding to Fe-In at 3.5Å. This does not mean that Fe-In bonds are formed in the ZISZ/Fe system, does it?
8. Figure 4e obtained from the UPS results show that the ZISZ/Fe possesses a gradient band structure throughout the nanosheets with the etching depth. The question here is whether the band gap of the system remains constant at different etching depths?
9. Finally, the authors should check the manuscript again, there are a number of errors in expressions. For example, when analyzing the supplementary Figure 8d, the peak position of O shifts toward a lower energy, rather than a higher energy. Moreover, there is an extra 'thick' in the part citing reference 31.

Reviewer #1 (Remarks to the Author):

The authors have answered most of the questions that I concern. Therefore, I consider the manuscript in this version can be accepted.

We thank the Reviewer for the constructive comments that helped us to improve the quality of our manuscript.

Reviewer #2 (Remarks to the Author):

The revised manuscript has carefully considered all the comments of the reviewers and made further improvement. I would suggest the acceptance and only have one minor comment on the the IPCE and APCE data of the ZISZ/Fe photoanodes in the supplementary Figure S13, why the data show some drop in the wavelength range of 360 nm.

[Author Reply]: Many thanks for your kind support and efforts in reviewing our manuscript. The drop in the wavelength range of 360 nm about the IPCE may be due to the reduced light utilization of photoanode in this wavelength. The similar phenomenon also happened in other literature, such as:

1. Zhang, H., et al. Gradient tantalum-doped hematite homojunction photoanode improves both photocurrents and turn on voltage for solar water splitting. *Nat. Commun.* **11**, 4622 (2020).
2. Lee, J., et al. Bendable BiVO₄-based photoanodes on a metal substrate realized through template engineering for photoelectrochemical water splitting. *ACS Appl. Mater. Interfaces* **13**, 16478-46484 (2021).
3. Zhang, S., Liu, Z., Chen, D., Yan, W., An efficient hole transfer pathway on hematite integrated by ultrathin Al₂O₃ interlayer and novel CuCoO_x cocatalyst for efficient photoelectrochemical water oxidation. *Appl. Catal. B Environ.* **277**, 119197 (2020).
4. Wu, Y., et al. Atomic Sandwiched p-n Homojunctions. *Angew. Chem. Int. Ed.* **60**, 3487-3492 (2021).

In addition, the APCE can be calculated from the formula:

$$APCE = \frac{IPCE}{\eta_{abs.}(\lambda)}$$

$\eta_{abs.}$ is the efficiency of the light harvested, and it can be calculated from the obtained light absorbance curves:

$$\eta_{\text{abs.}} = (1 - 10^{-A}) \times 100\%$$

where the A is light absorbance measured by a UV-vis spectroscopy.

Thus, owing to the similar rule of the UV-visible absorption spectra, the APCE and IPCE show the same trend.

Reviewer #3 (Remarks to the Author):

This authors have synthesized ordered nanosheet arrays ZISZ/Fe photoanode material by ALD, which shows reasonably high OER activity. DFT calculation was used to study the OER mechanism to understand and verify the experimental results. However, I find some arguments are rather vague, and conclusions are not fully supported their results, and suggest that the authors carefully consider the following issues:

We thank the Reviewer for the critical and constructive comments that helped us to improve the quality of our manuscript.

1. Hexagonal ZnIn_2S_4 is a layered material with the thickness of each layer about 10\AA , and the layers are connected by van der Waals interaction. When constructing the ZIS model for DFT calculation, why is the one-layer model used instead of the multi-layer model?

[Author Reply]: We thank the reviewer for raising this issue. As the Reviewer pointed out, the hexagonal ZnIn_2S_4 has a sandwiched layered structure, and the layers are connected by van der Waals interaction. The interlayer interaction is rather weak that allows one to use a monolayer structure instead of a multi-layer model in the simulations¹. Such settings not only maintain the major properties of the ZnIn_2S_4 because the catalytic reactions generally occur on the materials surface, but also reduce the computational cost. According to previous theoretical work¹⁻³, we constructed a 112-atom 4×4 monolayer ZnIn_2S_4 (001) slab with 64 S atoms, 16 Zn atoms and 32 In atoms to model ZIS system. We have added this discussion in the section of “Computational Details” during the revision.

1. Shi, X., Mao, L., Yang, P., Zheng, H., Fujitsuka, M., Zhang, J., Majima, T., Ultrathin

- ZnIn₂S₄ nanosheets with active (110) facet exposure and efficient charge separation for cocatalyst free photocatalytic hydrogen evolution. *Appl. Catal. B Environ.* **265**, 118616 (2020).
2. Luo, D., et al. Highly efficient photocatalytic water splitting utilizing a WO_{3-x}/ZnIn₂S₄ ultrathin nanosheet Zscheme catalyst. *J. Mater. Chem. A* **9**, 908-914 (2021).
 3. Zhang, Z., et al. Reasonably introduced ZnIn₂S₄@C to mediate polysulfide redox for long-life lithium–sulfur batteries. *ACS Appl. Mater. Int.* **13**, 141169-14180 (2021).

The following part has been incorporated into the revised manuscript.

DFT-D3 method of Grimme is employed to describe the van der Waals interactions⁵⁰. The dipole correction is taken into account for all calculations. The stoichiometric 4 × 4 one-layer 112-atom ZIS (001) surface with 64 S atoms, 16 Zn atoms, and 32 In atoms is built using the ZnIn₂S₄ unit cell⁵¹ based on previous DFT calculations⁵²⁻⁵⁴, because the hexagonal ZnIn₂S₄ has a sandwiched layered structure, and the layers are connected by van der Waals interaction. The interlayer interaction is rather weak that allows one to use a monolayer structure instead of a multi-layer model in the simulations⁵². Such settings not only maintain the major properties of the ZnIn₂S₄ because the catalytic reactions generally occur on the materials surface, but also reduce the computational cost.

50. Grimme, S., Antony, J., Ehrlich, S., Krieg, H. A consistent and accurate ab initio parametrization of density functional dispersion correction (DFT-D) for the 94 elements H-Pu. *J. Chem. Phys.* **132**, 154104 (2010).
51. Hu, X., Yu, J., Gong, J., Li, Q., Rapid mass production of hierarchically porous ZnIn₂S₄ submicrospheres via a microwave-solvothermal process. *Cryst. Growth Des.* **7**, 2444-2448 (2007).

52. Shi, X., Mao, L., Yang, P., Zheng, H., Fujitsuka, M., Zhang, J., Majima, T., Ultrathin ZnIn₂S₄ nanosheets with active (110) facet exposure and efficient charge separation for cocatalyst free photocatalytic hydrogen evolution. *Appl. Catal. B Environ.* **265**, 118616 (2020).
53. Luo, D., et al. Highly efficient photocatalytic water splitting utilizing a WO_{3-x}/ZnIn₂S₄ ultrathin nanosheet Zscheme catalyst. *J. Mater. Chem. A* **9**, 908-914 (2021).
54. Zhang, Z., et al. Reasonably introduced ZnIn₂S₄@C to mediate polysulfide redox for long-life lithium–sulfur batteries. *ACS Appl. Mater. Int.* **13**, 141169-14180 (2021).

2. The DFT model of ZISZ is stoichiometric, but is not without question. This model should be validated so as to be representative of the experimentally synthesized materials.

[Author Reply]: This is a good point to discuss, thank you. Our experimental measurements demonstrate that the ratio of elements Zn, In, and S is 10:16:34, indicating the ZISZ is stoichiometric. Thus, we have added an 8-atom Zn₄S₄ cluster on the ZnIn₂S₄ (001) surface, containing 16 Zn atom, 32 In atoms, and 64 S atoms, Supplementary Fig. 7a. The number of Zn, In, and S atoms in the ZISZ system are 20, 32, and 68, corresponding to a ratio of 10: 16: 34 and consistent with the present experiment. Shown by previous experimental and theoretical joint works on ZIS system, stoichiometric models are usually adopted for describing the experimental samples, such as WO_{3-x}/ZnIn₂S₄¹ and Li₂S_x/ZnIn₂S₄². In order to test the stability of our models, we have performed 8 picoseconds adiabatic molecular dynamics simulations at 300 K in the microcanonical ensemble at 300 K. The evolution of total energy cumulative average over time, Supplementary Fig. 19, shows small fluctuations around the equilibrium positions, demonstrating that all the ZISZ and ZISZ/Fe geometries are stable, Supplementary Fig. 7. We have added the corresponding description in the section of “Computational Details”

during the revision.

1. Luo, D., et al. Highly efficient photocatalytic water splitting utilizing a $\text{WO}_{3-x}/\text{ZnIn}_2\text{S}_4$ ultrathin nanosheet Zscheme catalyst. *J. Mater. Chem. A* **9**, 908-914 (2021).
2. Zhang, Z., et al. Reasonably introduced $\text{ZnIn}_2\text{S}_4@C$ to mediate polysulfide redox for long-life lithium–sulfur batteries. *ACS Appl. Mater. Int.* **13**, 141169-14180 (2021).

The following part has been incorporated into the revised manuscript.

According to the previous DFT calculations on the $\text{WO}_{3-x}/\text{ZnIn}_2\text{S}_4$ ⁵³ and the $\text{Li}_2\text{S}_x/\text{ZnIn}_2\text{S}_4$ systems, we add an 8-atom ZnS cluster into the ZIS (001) surface and leads to the stoichiometric ZISZ system, Supplementary Fig. 7a, consistent with present experimental conditions. The ZISZ/Fe system is obtained by replacing an In atom or a Zn atom with an Fe atom on the topmost ZISZ surface, displayed in Supplementary Fig. 7b-e. To eliminate the interactions between periodic images, a vacuum region of 20 Å is added normal to the surface.

Supplementary Figure 7. Models for formation energy calculations. Simulation cell showing the optimized geometries of (a) pristine ZISZ (001) surface and ZISZ/Fe structures of Fe substituting to (b, c) Zn and (d, e) In. The formation energies (E_f) listed below each geometry (b-e) are obtained using the formula: $E_f = E_{\text{ZISZ/Fe}} - E_{\text{ZISZ}} - E_{\text{Fe}} + E_{\text{Zn(In)}}$. Here,

$E_{\text{ZISZ/Fe}}$ and E_{ZISZ} are the total energy of the ZISZ/Fe and ZISZ systems, and E_{Fe} and $E_{\text{Zn(In)}}$ correspond to the energy of a single Fe, and Zn or In atom, calculated using the unit cell of each bulk material. Greater negative value of formation energy means easier substitution. Correspondingly, the estimation of OER efficiencies between pristine ZISZ and pristine ZISZ doping with Fe are determined by the systems (a) and (e).

Supplementary Figure 19. Evolution of total energy cumulative average over time for four systems oscillates only in a narrow window. Evolution of total energy of the 8 ps adiabatic MD trajectories in the microcanonical ensemble of pristine ZISZ (a) and (b-d) ZISZ/Fe systems. Small energy fluctuations indicate that the four systems used in the present work are thermodynamically stable.

3. In the DFT model of ZISZ/Fe, how to determine the position of the replaced In atom when replacing In with Fe, and whether is the replacement of In in the middle layer considered? The authors have mentioned that the experimentally synthesized ZISZ/Fe has Zn-O bonds at the bottom, but the role of O is not clearly explained. Also, have the authors considered the influence of Zn-O bond in the DFT calculation?

[Author Reply]: This is a good point, thank you. We calculated the formation energy of the replacement of In with Fe in the middle layer in the ZISZ system, Supplementary Fig. 7d, and found that this value (-4.92 eV) is 0.8 eV larger than the configuration when the Fe substitutes to one In atom at the bottom layer (-5.71 eV), Supplementary Fig. 7e. Therefore, we chose the geometry shown in Supplementary Fig. 7e to perform the OER calculations because the formation energies of the two configurations by replacing of Zn with Fe are also larger than the case shown in Supplementary Fig. 7e. As a result, the original discussion and results remain unchanged (Fig. 3e and f).

In the DFT calculation, we not considered the influence of Zn-O bond, the reason as follows:

1) In the experimental part, in order to prove that the promotion of surface OER is not related to O, only O₃ was introduced into the ALD chamber (samples are denoted as ZISZ/O) to avoid the formation of Fe-In-S. The *J* of ZISZ/O is lower (Supplementary Fig. 17a) and the surface resistance is higher (Supplementary Fig. 17b), therefore, the Fe-In-S clusters are critical for enhanced OER reaction.

2) Many literatures have reported the role of O doping, we can find that O doping is not always advantageous. For the real role of O doping, a lot of works are reported through DFT. The amount of O is critical, only proper content of O doping can promote OER. In this manuscript, the presence of O does not promote surface OER, as shown in Supplementary Fig. 17. The main function of O in here is to form a gradient band, which further promotes the bulk transport of carriers. Thus, we do not consider the influence of Zn-O bond during the DFT calculation. Some references about the O doping can illustrate these:

1. Yu, Y., Huang Y., Yu, Y., Shi, Y., Zhang, B. Design of continuous built-in band bending in

- self-supported CdS nanorod-based hierarchical architecture for efficient photoelectrochemical hydrogen production. *Nano Energy*, **43**, 236-243 (2018).
2. Hou, J., et al. Atomically thin mesoporous In₂O_{3-x}/In₂S₃ lateral heterostructures enabling robust broadband-light photo-electrochemical water splitting. *Adv. Energy Mater.* **8**, 1701114 (2017).
 3. Tian, W., Chen, C., Meng, L., Xu, W., Cao, F., Li, L. PVP treatment induced gradient oxygen doping in In₂S₃ nanosheet to boost solar water oxidation of WO₃ nanoarray photoanode. *Adv. Energy Mater.* **10**, 1903951 (2020).
 4. Yang, W., et al. Enhanced photoexcited carrier separation in oxygen-doped ZnIn₂S₄ nanosheets for hydrogen evolution. *Angew. Chem. Int. Ed.* **55**, 6716-6720 (2016).
 5. Jiao, X., et al. Partially oxidized SnS₂ atomic layers achieving efficient visible-light-driven CO₂ reduction. *J. Am. Chem. Soc.* **139**, 18044-18051 (2017).
 6. Meng, L., Rao, D., Tian, W., Cao, F., Yan, X., Li, L. Simultaneous manipulation of O-doping and metal vacancy in atomically thin Zn₁₀In₁₆S₃₄ nanosheet arrays toward improved photoelectrochemical performance. *Angew. Chem. Int. Ed.* **57**, 16882-16887 (2018).

The following part has been incorporated into the revised manuscript.

To rationalize the improved catalytic performance of the ZISZ/Fe photoanode, we obtained the η for OER of the ZISZ and ZISZ/Fe systems based on the DFT calculated free energy changes (Supplementary Table 2) using the two structures shown in Supplementary Fig. 7a and e. The computational details are described in Supplementary Material. The η decreases to 2.499

V in the ZISZ/Fe from 3.522 V in the pristine ZISZ (Fig. 3e and f). The notable 1.023 V drop in η provides a solid evidence that the catalytic performance is enhanced in the ZISZ/Fe photoanode once the Fe-In-S clusters are formed, because the Fe dopant alters the rate-determining step of $*O$ to $*OOH$ (step 3) in the ZISZ system (Fig. 3e) into $*OH$ to $*O$ (step 2) in the ZISZ/Fe (Fig. 3f) system, leading to that the OOH^* is preferred for formation on the ZISZ/Fe catalyst and favors O_2 production.

Fig. 3. PEC performance, optical, and electrochemical characterizations. a Linear sweep

voltammogram curves. **b** UV-visible absorption spectra (the inset is the corresponding Tauc-plots). **c** The η_{sep} and **(d)** the η_{inj} of ZIS, ZISZ, and ZISZ/Fe. Free energies of OER reaction steps for **(e)** ZISZ and **(f)** ZISZ/Fe systems, the entire structure of the OER four-step reaction are shown in Supplementary Fig. 19.

Supplementary Figure 7. Models for formation energy calculations. Simulation cell showing the optimized geometries of **(a)** pristine ZISZ (001) surface and ZISZ/Fe structures of Fe substituting to **(b, c)** Zn and **(d, e)** In. The formation energies (E_f) listed below each geometry (b-e) are obtained using the formula: $E_f = E_{\text{ZISZ/Fe}} - E_{\text{ZISZ}} - E_{\text{Fe}} + E_{\text{Zn(In)}}$. Here, $E_{\text{ZISZ/Fe}}$ and E_{ZISZ} are the total energy of the ZISZ/Fe and ZISZ systems, and E_{Fe} and $E_{\text{Zn(In)}}$ correspond to the energy of a single Fe, and Zn or In atom, calculated using the unit cell of each bulk material. Greater negative value of formation energy means easier substitution. Correspondingly, the estimation of OER efficiencies between pristine ZISZ and pristine ZISZ doping with Fe are determined by the systems **(a)** and **(e)**.

Supplementary Figure 17. Experiment comparison. (a) Linear sweep voltammogram curves, (b) EIS plots of ZISZ/O (only O₃) and ZISZ/Fe.

The following part has been incorporated into the revised manuscript.

Supplementary Table 2. The free energy changes of four elementary steps for OER in ZISZ, and ZISZ/Fe systems when the applied potential is 0 V or 1.23 V, and the unit of free energy is eV.

	U (V)	step1	step2	step3	step4
ZISZ	0	0.536	0.047	3.522	0.815
	1.23	-0.694	-1.183	2.292	-0.415
ZISZ/Fe	0	0.833	2.499	0.071	1.517
	1.23	-0.397	1.269	-1.159	0.287

4. In these DFT models, vacuum is added to eliminate the influence of periodic boundary conditions, but for these asymmetric models, is the dipole correction considered in the calculation? In addition, what are the oxidation state and spin state of Fe introduced in the system?

[Author Reply]: We thank the Reviewer for pointing this out. We have repeated all calculations with dipole correction. The results show that the values of formation energy, free energy of each step for OER, and the evolution of total energy for MD simulations differ from the data obtained without dipole corrections. However, the conclusions derived from the original calculations remain unchanged. Thus, we simply replaced the old data with new ones in the revised manuscript. In order to determine the oxidation state of Fe in the ZISZ/Fe system, we calculated the Bader charge and which is 1.687 on the Fe dopant, corresponding to +2 oxidation state. The calculated magnetic moment of Fe is 2.662, indicating the spin state of Fe dopant is 3. We have added the corresponding discussion in the sections of “Computational Details” and in the 2nd paragraph of section “Results and discussion” during the revision.

The following part has been incorporated into the revised manuscript.

The calculated Bader charge and magnetic moment on the Fe ion correspond to 1.687 and 2.662, indicating that the oxidation state and spin state of the Fe in the ZISZ/Fe system are +2 and 3.

5. The entire structure of the OER four-step reaction calculated by DFT should be given in detail in the supplementary information, rather than just some partial structures in Figure 3e and Figure 3f. Moreover, the original total energy given in Supplementary Figure 19 can be converted into a cumulative average over time, will it be clearer?

[Author Reply]: As the Reviewer required, we have provided all structures of the OER four-step reaction calculated by DFT in Supplementary Fig. 20. The original total energy given in Supplementary Fig. 19 has been converted into a cumulative average over time. The small energy fluctuations around the equilibrium positions demonstrate that all structures are stable in the present study. We have added the corresponding description in the section of “Computational Details” during the revision.

The following part has been incorporated into the revised manuscript.

Supplementary Figure 19. Evolution of total energy cumulative average over time for four systems oscillates only in a narrow window. Evolution of total energy of the 8 ps adiabatic MD trajectories in the microcanonical ensemble of pristine ZISZ (a) and (b-d) ZISZ/Fe systems. Small energy fluctuations indicate that the four systems used in the present work are thermodynamically stable.

Supplementary Figure 20. All the optimized structures of the OER four-step reaction for both the ZISZ and ZISZ/Fe systems.

6. Figure 2d and Supplementary Figure 6b analyze Fe 2p through XPS spectroscopy. As the number of cycles increases, the peak position and chemical composition of Fe have changed. How do you explain the change in chemical composition here?

[Author Reply]: Many thanks for your kind support and efforts in reviewing our manuscript.

According to the data in this manuscript, as the number of cycles increases, the peak position of Fe has changed. The peaks located at 712.92 eV and 710.86 eV correspond to Fe³⁺ and Fe²⁺, respectively, indicating the chemical state of substituted Fe changes with increased Fe. In other words, with the appearance of a new peak of Fe in XPS, the performance of photoanode shows a downward trend. As we discussed in this manuscript, with the increased cycles, the peak intensity of In gradually decreases while that of Fe increases. At the beginning, the In³⁺ was replaced by Fe³⁺, and with the increased Fe, the extra Fe may react with Fe³⁺, which caused the appearance of Fe²⁺. The new chemical composition act as defect state, seriously affect the performance of the device.

7. Figure 2f is obtained by Fourier transform of the Fe k-edge EXAFS. As the number of ALD cycles increases, no obvious peak is seen at the position corresponding to Fe-In at 3.5Å. This does not mean that Fe-In bonds are formed in the ZISZ/Fe system, does it?

[Author Reply]: Many thanks for your suggestions in reviewing our manuscript. This phenomenon does not mean that Fe-In bonds are not formed in the ZISZ/Fe system.

This fitting shows the presence of In in structure in term of Fe-In and Fe-S-In which contributes the r-space signal of the fit. Additionally, the distance between 3.5-4.5 angstrom is occupied by Fe-In and Fe-S-In signals as seen in the below figure.

Fig. R1. The FT-EXAFS of ZISZ/Fe and fit curves according to the standard model.

8. Figure 4e obtained from the UPS results show that the ZISZ/Fe possesses a gradient band structure throughout the nanosheets with the etching depth. The question here is whether the band gap of the system remains constant at different etching depths?

[Author Reply]: Many thanks for your suggestions in reviewing our manuscript. As you said, the band gap of the system should not remain constant at different etching depths. Nevertheless, the band gap of ZISZ and ZISZ/Fe are slightly different in this manuscript, which the difference is only 0.04 eV. Thus, the band gap in the different etching depths of ZISZ/Fe is not very different so as to affect the direction of carrier movement.

In other words, O-doped composition appeared at the bottom of the nanosheet. According to some references, the O doping can cause the band gap to narrow^[1-3]. Therefore, the bottom of the sample should be a material with a narrower band gap. According to this statement, the direction of carrier movement remains constant.

On the other hand, limited by current technical means, the area during the Ar etching process is too small to be detected. Thus, the band gap of different etching depth cannot be detected. Some

literature also employed similar means for conducting this test, such as considering band gap remains unchanged. The relevant documents are as follows^[4-8]:

1. Hou, J., et al. Atomically thin mesoporous $\text{In}_2\text{O}_{3-x}/\text{In}_2\text{S}_3$ lateral heterostructures enabling robust broadband-light photo-electrochemical water splitting *Adv. Energy Mater.* **17**, 1701114 (2017).
2. Yang, W., et al. Enhanced photoexcited carrier separation in oxygen-doped ZnIn_2S_4 nanosheets for hydrogen evolution. *Angew. Chem. Int. Ed.* **55**, 6716-6720 (2016).
3. Jiao, X., et al. Partially oxidized SnS_2 atomic layers achieving efficient visible-light-driven CO_2 reduction. *J. Am. Chem. Soc.* **139**, 18044-18051 (2017).
4. Yu, Y., Huang, Y., Yu, Y., Shi, Y., Zhang, B. Design of continuous built-in band bending in self-supported CdS nanorod-based hierarchical architecture for efficient photoelectrochemical hydrogen production. *Nano Energy* **43**, 236-243 (2018).
5. Luo, Z., Li, C., Liu, S., Wang, T., Gong, J., Gradient doping of phosphorus in Fe_2O_3 nanoarray photoanodes for enhanced charge separation. *Chem. Sci.*, **8**, 91-100 (2017).
6. Kong, D., et al., Ti-gradient doping to stabilize layered surface structure for high performance high-Ni oxide cathode of Li-Ion battery. *Adv. Energy Mater.*, **9**, 1901756 (2019).
7. Zhao, Y., et al., surface structural transition induced by gradient polyanion-doping in Li-rich layered oxides: implications for enhanced electrochemical performance. *Adv. Funct. Mater.*, **26**, 4760-4767 (2016).
8. Huang, H., et al., Oriented built-in electric field introduced by surface gradient diffusion doping for enhanced photocatalytic H_2 evolution in CdS Nanorods. *Nano Lett.*, **17**, 3803-

3808 (2017).

9. Finally, the authors should check the manuscript again, there are a number of errors in expressions. For example, when analyzing the supplementary Figure 8d, the peak position of O shifts toward a lower energy, rather than a higher energy. Moreover, there is an extra 'thick' in the part citing reference 31.

[Author Reply]: Many thanks for your suggestions and efforts in reviewing our manuscript. we have checked and polished the manuscript.

The following part has been incorporated into the revised manuscript.

These cocatalysts may lead to the decreased light harvesting capability and increased charge recombination rate of photoanodes, owing to the thickness of cocatalyst layer and additional interface defects introduced between cocatalysts and photoanodes³¹.

When the etching depth increases to 50 nm, the signal of Fe substantially disappears (Supplementary Fig. 8c), but the peak position of O shifts toward lower energy (Supplementary Fig. 8d).

REVIEWER COMMENTS

Reviewer #3 (Remarks to the Author):

This reviewer is happy to see that the authors have addressed the issues raised, and therefore recommends its publication in nature common.